# WOODS: Benchmarks for Out-of-Distribution Generalization in Time Series

**Jean-Christophe Gagnon-Audet** *jean-christophe.gagnon-audet@mila.quebec*
*Mila - Québec AI Institute*
*University of Montreal*

**Kartik Ahuja** *kartik.ahuja@mila.quebec*
*Mila - Québec AI Institute*
*University of Montreal*

**Mohammad-Javad Darvishi-Bayazi** *mohammad.bayazi@mila.quebec*
*Mila - Québec AI Institute*
*University of Montreal*

**Pooneh Mousavi** *mousavi.pooneh@gmail.com*
*Gina Cody School of Engineering and Computer Science*
*Concordia University*

**Guillaume Dumas** *guillaume.dumas@ppsp.team*
*Mila - Québec AI Institute*
*CHU Sainte-Justine Research Center, Department of Psychiatry*
*University of Montreal*

**Irina Rish** *irina.rish@mila.quebec*
*Mila - Québec AI Institute*
*University of Montreal*

**Reviewed on OpenReview:** *https://openreview.net/forum?id=mvftzofTYQ*

## Abstract

Deep learning models often fail to generalize well under distribution shifts. Understanding and overcoming these failures have led to a new research field on Out-of-Distribution (OOD) generalization. Despite being extensively studied for static computer vision tasks, OOD generalization has been severely underexplored for time series tasks. To shine a light on this gap, we present WOODS: 11 challenging time series benchmarks covering a diverse range of data modalities, such as videos, brain recordings, and smart device sensory signals. We revise the existing OOD generalization algorithms for time series tasks and evaluate them using our systematic framework. Our experiments show a large room for improvement for empirical risk minimization and OOD generalization algorithms on our datasets, thus underscoring the new challenges posed by time series tasks.

# 1 Introduction

In the last decade, the success of deep learning has led to impactful applications spanning many fields (Krizhevsky et al., 2012; Vaswani et al., 2017b; Silver et al., 2016; Jumper et al., 2021; Brown et al., 2020b). However, parallel to this surge, there is growing evidence that deep learning models exploit undesired correlations due to selection biases, confounding factors, and other biases in the data (Geirhos et al., 2020; Shen et al., 2021; Ye et al., 2021a). These biases can often create shortcuts that help the model arrive at low empirical risk on a dataset. Nevertheless, a prediction rule relying on these shortcuts will not generalize out of its training distribution as it uses spuriously correlated factors instead of causal factors (Rojas-Carulla et al., 2015; Schölkopf et al., 2021). Such a failure becomes very concerning in real-life applications that directly impact human lives, such as medicine (Razzak et al., 2018; Ching et al., 2018; Rajkomar et al., 2018) or self-driving cars (Badue et al., 2021; Janai et al., 2020).

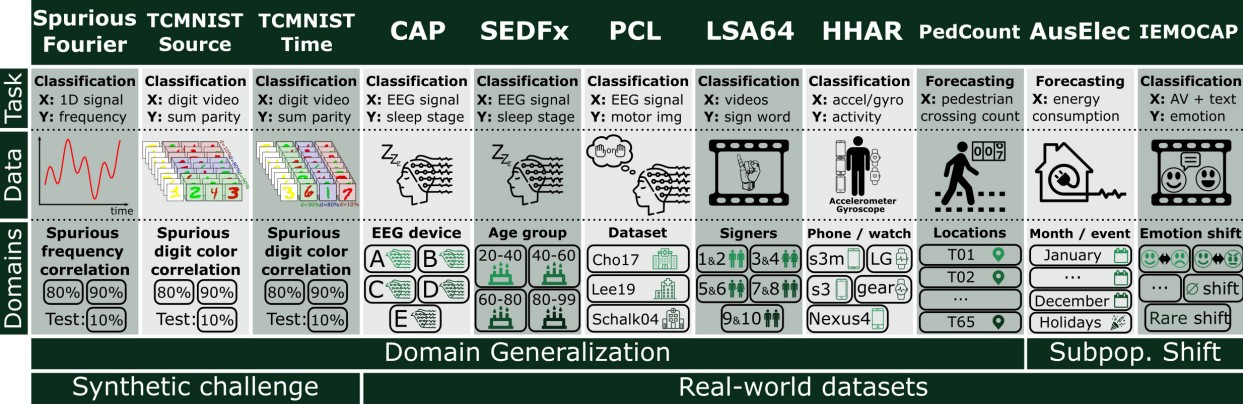

Figure 1: Summary of WOODS benchmark: tasks, modalities, domains and distribution shifts.

Let us explain an important failure mode with a common example from the work of Beery et al. (2018). Consider the task of distinguishing cows and camels in pictures. The training dataset is heavily tainted by selection bias, as the vast majority of cow images were taken in green pastures, and the vast majority of camel images were taken in sandy areas. A model trained to minimize empirical risk over the training dataset leverages the selection bias and ends up using green background to classify cows and beige backgrounds to classify camels. As a way to capture different failures of deep learning models, much work has gone into finding and standardizing datasets with distribution shifts (Gulrajani & Lopez-Paz, 2020; Ye et al., 2021b; Koh et al., 2021). These datasets provide a direction for research efforts in the field of OOD generalization. Gulrajani & Lopez-Paz (2020) gathered seven standard image datasets with distribution shifts and concluded that no OOD generalization algorithm considerably outperformed ERM, highlighting the need for better and more versatile solutions. Ye et al. (2021b) showed that some algorithms outperform ERM on specific types of shifts, highlighting that different algorithms might be needed for different type of distribution shifts. Koh et al. (2021) created a set of benchmarks of in-the-wild distribution shifts, highlighting the challenges in real-world applications. Further related works can be found in Appendix B.

The above mentioned works have led to crucial empirical and theoretical insights towards addressing the OOD generalization failure in deep learning. However, they have been predominantly focused on static computer vision tasks, leaving the field of time series severely underexplored despite being essential to various applications such as computational medicine (Topol, 2019; Yang et al., 2021; Jarrett et al., 2021), natural sciences (Stoffer & Ombao, 2012; Tanaka et al., 2021), finance (Sezer et al., 2020; Heaton et al., 2016; Andersen et al., 2005), climate (Mudelsee, 2019), retail (Böse et al., 2017), ecology (Capinha et al., 2021; Christin et al., 2019), energy (Deb et al., 2017) and many more (Torres et al., 2021; Lim & Zohren, 2021). In this work, we take the first step towards a deeper understanding of distribution shifts in time series data. Our key contributions are:

- We propose WOODS: a benchmark of 3 synthetic challenge and 8 real-world datasets, totaling 11 datasets spanning a wide array of critical problems and data modalities, such as videos, brain recordings, and smart device sensory signals (See Figure 1).

- We develop a systematic framework for easy evaluation of new time series datasets and algorithms. The framework includes adaptation of existing OOD generalization algorithms for time series datasets.

- We conduct extensive experiments on the above datasets with ERM and various OOD generalization algorithms. Our findings lead us to conclude that OOD generalization in time series brings its own set of challenges and that there is a large room for improvement as shown in Table 1.

**Why OOD generalization in time series?** Recently, work in the deep learning community have shown that large scale pretrained models such as CLIP (Radford et al., 2021a) show considerable improvements when it comes to OOD generalization performance for static computer vision tasks (Cha et al., 2022). Since large scale pretrained models for time series data do not exist yet, whether or not large scale pretraining on time series data helps address OOD generalization challenge of time series remains to be determined. We hope our datasets and benchmarks help shed light on this important question.

In the next section, we discuss problem formulation, followed by discussion on the various datasets we use. In Section 5, we describe the adaptation of existing methods for time series settings. In Section 6, we discuss the results followed by the conclusion and limitations in Section 7.

Table 1: Generalization gap between the In-Distribution (ID) performance and the OOD performance of ERM on the WOODS benchmarks. See Section 6.2 for more details.

| Dataset | Performance | | |
|---|---|---|---|
| (Perf. is accuracy unless specified) | ID | OOD | **Gap** |
| Spur.-Fourier | 74.5 (0.1) | 9.8 (0.2) | 64.7 |
| TCM.-Source | 68.4 (0.1) | 10.2 (0.1) | 58.2 |
| TCM.-Time | 89.4 (0.0) | 10.0 (0.0) | 79.3 |
| CAP | 75.1 (0.7) | 62.8 (0.6) | 12.3 |
| SEDFx | 72.5 (0.4) | 67.3 (0.8) | 5.2 |
| PCL | 73.6 (0.2) | 64.3 (0.5) | 9.3 |
| HHAR | 93.4 (0.4) | 84.4 (0.6) | 9.0 |
| LSA64 | 86.6 (1.0) | 53.4 (2.0) | 33.2 |
| PedCount (rmse) | 99.1 (2.7) | 204.9 (11.4) | 105.8 |
| AusElec (rmse) | 232.0 (2.6) | 397.2 (8.4) | 165.2 |
| IEMOCAP | 69.1 (0.4) | 57.7 (1.9) | 11.4 |

## 2 Problem formulation

### 2.1 Static tasks

Consider the standard OOD generalization setting for static supervised learning tasks. Data samples: $(X, Y)$ consists of the input observation $X$ and the corresponding label $Y$. We gather the datasets $D^d$ from the domains $d \in \mathcal{E}^{\text{train}}$ which follow the follows the distribution $\mathbb{P}^d(X, Y)$. Datasets from these domains form the training dataset $D^{\text{train}}$ which follows the training distribution $\mathbb{P}^{\text{train}} = \sum_{d \in \mathcal{E}^{\text{train}}} q_d^{\text{train}} \mathbb{P}^d$, where $q^{\text{train}} \in \mathbb{R}^{|\mathcal{E}^{\text{train}}|}$ is the vector of training mixture weights and $q_d^{\text{train}}$ is the mixture weight for domain $d$. We define a predictor $f$. The performance of $f$ on domain $d$ is measured in terms of the risk $R^d(f) = \mathbb{E}^d\big[\ell(f(X), Y)\big]$, where $\mathbb{E}^d$ is the expectation over the distribution $\mathbb{P}^d$ and $\ell \to \mathbb{R}_{\geq 0}$ denotes the loss function. We evaluate the predictor on a set of test domains denoted as $\mathcal{E}^{\text{all}}$. The goal of OOD generalization is to use the training dataset $D^{\text{train}}$ and construct a predictor $f$ that can perform well on the test domains. We write this objective formally below.

**Problem 2.1.** Find a predictor $f^*$ that solves $\min_f \max_{d \in \mathcal{E}^{\text{all}}} R^d(f)$.

In the above problem, some restrictions are necessary on the set of testing domains $\mathcal{E}^{\text{all}}$ to make Problem 2.1 of practical interest. Otherwise, the best predictor is random guessing, as nothing can be assumed about the test domains. Many works (Arjovsky et al., 2020; Chen et al., 2021; Sun & Saenko, 2016) provide guarantees of generalizing to OOD domains by assuming that the relationship between the label and some subset of features (potentially a nonlinear transform of the observation (Rojas-Carulla et al., 2018; Ahuja et al., 2020b)) *is invariant across all domains*. We call this subset of features the *invariant features*, and any other features that might be correlated with the label are called *spurious features*. The predictor $f^*$ solving Problem 2.1 is said to be *OOD-optimal*; $f^*$ relies on the invariant features that generalize to all domains in $\mathcal{E}^{\text{all}}$ (Koyama &

Yamaguchi, 2020). Because the set of training domains $\mathcal{E}^{\text{train}}$ is much smaller than the set of testing domains $\mathcal{E}^{\text{all}}$, learning features that generalize to all test domains is a challenging task.

In practice, we aim to solve Problem 2.1 to avoid the predictor to fail at test time when evaluated on the test dataset $D^{\text{test}}$ which follows the test distribution $\mathbb{P}^{\text{test}} = \sum_{d \in \mathcal{E}^{\text{test}}} q_d^{\text{test}} \mathbb{P}^d$, where $q^{\text{test}} \in \mathbb{R}^{|\mathcal{E}^{\text{test}}|}$ is the vector of training mixture weights and $q_d^{\text{test}}$ is the mixture weight for domain $d$. There exists 2 significant ways the distribution $\mathbb{P}^{\text{test}}$ can shift:

- **Domain generalization** (Arjovsky et al., 2020) The test domains are a superset of the training domains, such that $\mathcal{E}^{\text{train}} \subseteq \mathcal{E}^{\text{test}}$. We seek to generalize to the unseen domains $\mathcal{E}^{\text{train}} \setminus \mathcal{E}^{\text{test}}$ (Gulrajani & Lopez-Paz, 2020; Wang et al., 2022).

- **Subpopulation shift** (Koh et al., 2021) There are no unseen domains, such that $\mathcal{E}^{\text{train}} \supseteq \mathcal{E}^{\text{test}}$, however, the test domains mixture is different than the training distribution such that $q_d^{\text{train}} \neq q_d^{\text{test}}$. We seek to minimize the maximum domain error in $\mathcal{E}^{\text{test}}$ (Sagawa et al., 2020; Yang et al., 2023; Santurkar et al., 2020).

## 2.2 Time series tasks

Data samples consist of the input time series observation $\mathbf{X} = [X_t]_{t \in S_t}$, where $S_t$ is the set of time steps, and the set of labels $\mathbf{Y} = [Y_t]_{t \in S_p}$, where $S_p \subseteq S_t$ is the set of labeled time steps. The performance of the predictor $f$ is measured in terms of the risk $R^d(f) = \mathbb{E}^d\big[\ell(f(\mathbf{X}), \mathbf{Y})\big]$, where the expectation is taken over time samples from domain $d$. We formalize the OOD generalization problem in time series as Problem 2.1.

In time series, similar to static tasks, the distribution shift can occur across data sources. Additionally, the distribution can also shift over time. As a concrete real-world example of this characteristic, consider a predictor monitoring a person's health from vital signs gathered with a smart watch.

**Example 2.2** (Source-domains). Wrist characteristics such as size or hair vary across person, or *sources*. The solution to Problem 2.1 with persons as source domains $d$ would be a predictor that does not rely on spurious wrist characteristics and thus generalizes to new persons. We call this formulation of domains as *Source-domains* as time series are taken from different sources, see Figure 2(b).

**Example 2.3** (Time-domains). Heart rate is lower during the night when we are asleep and higher during the day when we are awake. However, when we are working during the night, our heart rate might be higher than on a typical night. A predictor that relies on spurious features like the time of day could make a false alarm regarding our health on an atypical day. The solution to Problem 2.1 with time of day as time domains $d$ would be a predictor that does not rely on spurious features, and thus generalizes to different activities at different times. We call this way of defining domains *Time-domains*, as the data distribution changes through time, see Figure 2(c).

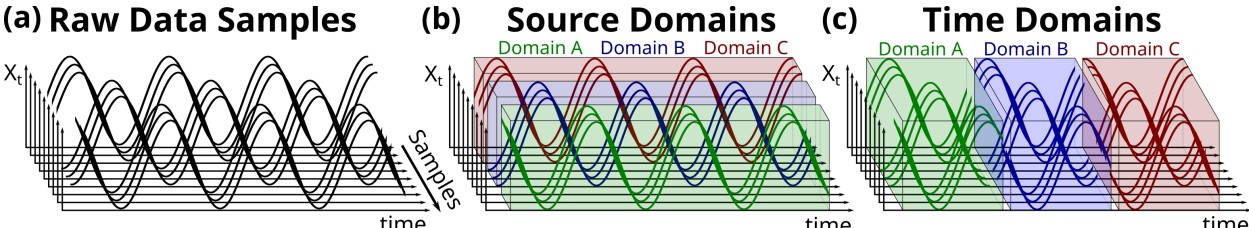

Figure 2: Illustration of the Source- and Time-domain definitions.

# 3 Synthetic challenge datasets

## 3.1 Spurious-Fourier: Spurious features encoded in the frequency domain

Colored MNIST (CMNIST) (Arjovsky et al., 2020) presented the failure mode of ERM under distribution shift in the image domain. This was accomplished by creating training domains with strongly predictive

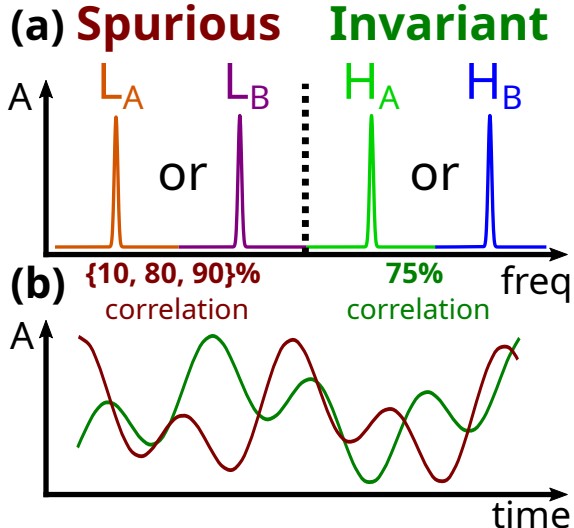

Figure 3: (a) Fourier spectrum construction in the Spurious-Fourier dataset. Signals have one low-frequency peak and one high-frequency peak. Signals are constructed from the Fourier spectrum with an inverse Fourier transform. (b) Examples of reconstructed signals, both signals have the same high frequency, but different low frequencies, which are hard to distinguish visually.

spurious features and weakly predictive invariant features. The spurious correlation would be flipped at test time while the invariant correlation was kept the same. The correlation flip made it clear which features the model relied on to make predictions.

We create a dataset composed of one-dimensional signals, where the task is to perform binary classification based on the frequency characteristics. Signals are constructed from Fourier spectra with one low-frequency peak ($L_{A \text{ or } B}$) and one high-frequency peak ($H_{A \text{ or } B}$), see Figure 3. Domains $D^d|_{d \in \{10\%, 80\%, 90\%\}}$ contain signal-label pairs, where the labels are created such that the information carried by the low-frequency signal are d% correlated with the label (varies by domain), while the information carried by the high-frequency signal is 75% correlated with the label.

In the training dataset $D^d|_{d \in \{80\%, 90\%\}}$, the low-frequency signal are a stronger predictor of the label (85%) than the high-frequency signal (75%). Therefore, minimizing the empirical risk fails at learning the invariant high frequencies as the low frequencies achieve the lower risk.

Appendix C.1 provides more information about the dataset.

### 3.2 Temporal Colored MNIST: A study of domain definitions in sequential data

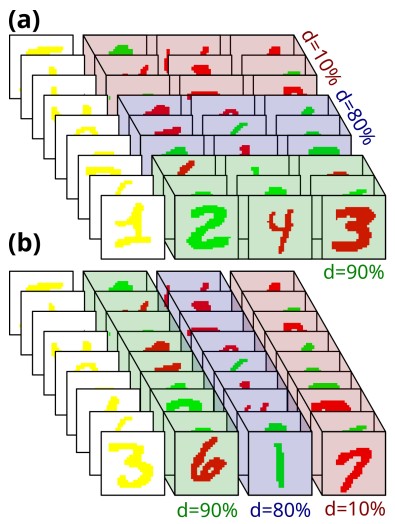

Figure 4: Domain definition of both TCMNIST (a) Source and (b) Time datasets. Data samples are videos of four colored MNIST digits where the task is to predict whether the sum of the current and previous digits in the sequence is odd or even. The color is spuriously correlated with the label.

In Temporal CMNIST (TCMNIST), we extend the CMNIST dataset to a binary classification task of video frames in order to investigate both domain definition paradigms presented in Section 2.2: Source-domains (Example 2.2) and Time-domains (Example 2.3). Videos are sequences of four colored MNIST digits where the goal is to predict whether the sum of the current and previous digits in the sequence is odd or even, see Figure 4. Prediction is made for all frames except for the first one. The labels are created such that the information carried by the color of the digits are d% correlated with the label (varies by domain), while the information carried by the value of the digit is 75% correlated with the label.

**TCMNIST-Source** Domains are created such that the color correlation is constant among the frames of a video, but varies between video from different domains $d \in \{10\%, 80\%, 90\%\}$. The domain definition is depicted in Figure 4(a).

Appendix C.2 provides more information about the dataset.

**TCMNIST-Time** Domains are created such that the color correlation varies across frames. However, videos all have the same sequence of color correlation, where the first labeled frame correlation is 90%, second is 80% and third is 10%. The domain definition is depicted in Figure 4(b).

Appendix C.3 provides more information about the dataset.

## 4 Real-world datasets

### 4.1 CAP: Sleep classification across different machines

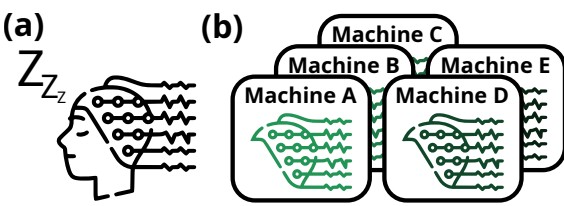

Figure 5: Summary of the CAP dataset. (a) The task is to perform sleep stage classification from EEG measurements. (b) The dataset has five source domains, where each domain contains data gathered with a different machine. The goal is to generalize to unseen machines.

A recurrent problem in computational medicine is that models trained on data from a given recording device will not generalize to data coming from another device, even when both devices are from a similar equipment provider. Failure to generalize to unseen machines can cause critical issues for clinical practice because a false sense of confidence in a model could lead to a false diagnosis (Kim et al., 2018; Engemann et al., 2018). We study these machinery-induced distribution shifts with the CAP (Terzano et al., 2001; Goldberger et al., 2000) dataset (Figure 5).

We consider the sleep stage classification task from electroencephalographic (EEG) measurements. The dataset has five source domains, where each domain contains data gathered with a different machine. The goal is to generalize to unseen machines.

Appendix C.4 provides more information about the dataset.

### 4.2 SEDFx: Sleep classification across age groups

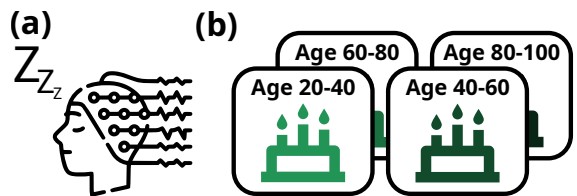

Figure 6: Summary of the SEDFx dataset. (a) The task is to perform sleep stage classification from EEG measurements. (b) The dataset has four source domains, where each domain contains data from participants of a certain age group. The goal is to generalize to unseen age groups.

In clinical settings, we train a model on the data gathered from a limited number of patients and hope this model will generalize to new patients in the future (Pfohl et al., 2022). However, this generalization between observed patients in the training dataset and new patients is not guaranteed. Distribution shifts caused by shifts in patient demographics (e.g., age, gender, and ethnicity) can cause the model to fail. We study age demographic shift with the SEDFx (Kemp et al., 2000; Goldberger et al., 2000) dataset (Figure 6).

We consider the sleep classification task from EEG measurements. The dataset has four source domains, where each domain contains data from participants of a certain age group. The goal is to generalize to unseen age groups.

Appendix C.5 provides more information about the dataset.

### 4.3 PCL: Motor imagery classification across data-gathering procedures

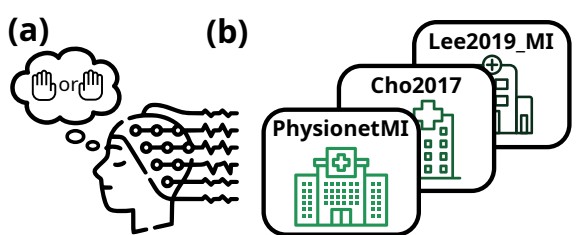

Figure 7: Summary of the PCL dataset. (a) The task is to perform motor imagery classification from EEG measurements. (b) The dataset has three source domains, where each domain contains a dataset from a different research group carrying out the same task. The goal is to generalize to unseen datasets of the same task.

Aside from changes in the recording device and shifts in patient demographics, human intervention in the data gathering process is another contributing factor to the distribution shift that can lead to failure of clinical models (e.g., Camelyon17 (Koh et al., 2021; Sagawa et al., 2021)). This challenge is especially prevalent in temporal medical data (e.g., EEG, MEG, and others) because recording devices are complex tools greatly affected by nonlinear effects and modulations. These effects are often caused by context and preparations made before the recording (Engemann et al., 2018). We study these procedural shifts with the PCL (Lee et al., 2019; Cho et al., 2017; Schalk et al., 2004; Jayaram & Barachant, 2018) dataset (Figure 7).

We consider the motor imagery task from EEG measurements. The dataset has three source domains, where each domain contains a dataset from a different research group carrying out the same task. The goal is to generalize to unseen datasets of the same task.

Appendix C.6 provides more information about the dataset.

### 4.4 LSA64: Sign language video classification across speakers

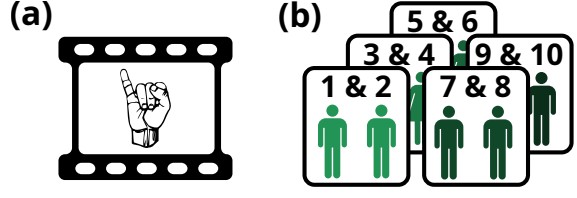

Figure 8: Summary of the LSA64 dataset. (a) The task is to perform signed word classification from videos. (b) The dataset has five source domains, where each domain contains videos of different signers. The goal is to generalize to unseen signers.

Communication is an individualistic way to convey information through different media: text, speech, body language, and many others. However, some media are more distinctive and challenging than others. For example, text communication has less inter-individual variability than body language or speech. If deep learning systems hope to interact with humans effectively, models need to generalize to new and evolving mannerisms, accents, and other subtle variations in communication that significantly impact the meaning of the message conveyed. We study the ability of models to recognize information coming from unseen individuals with the LSA64 (Ronchetti et al., 2016) dataset (Figure 8).

We consider the video classification of signed words in Argentinian Sign Language. The dataset has five source domains, where each domain contains videos of different signers. The goal is to generalize to unseen signers.

Appendix C.7 provides more information about the dataset.

## 4.5 HHAR: Human activity recognition across smart devices

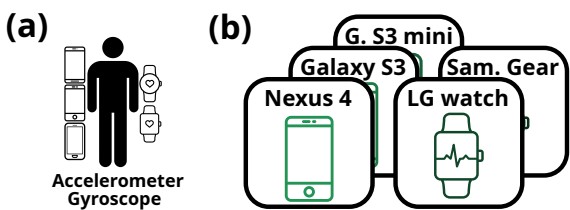

Figure 9: Summary of the HHAR dataset. (a) The task is to perform human activity classification from smart devices sensory data. (b) The dataset has five source domains, where each domain contains data gathered with a different smart device. The goal is to generalize to unseen smart devices.

The intrinsic biases from inaccurate and poorly calibrated sensors of smart devices, along with the accumulated biases from everyday use makes human activity recognition a notoriously difficult task when task when done across devices (Stisen et al., 2015; Blunck et al., 2013). Contrary to static tasks where uninformative features can often be segmented out from the input features (e.g., background when classifying an animal from an image), invariant features in time series are often highly convoluted with other spurious features. We study the ability of models to ignore spurious information from complex signals with the HHAR (Stisen et al., 2015; Dua & Graff, 2017) dataset (Figure 9).

We consider the human activity classification task from accelerometer and gyroscope measurements of smartphones and smartwatches. The dataset has five source domains, where each domain contains data gathered with a different device. The goal is to generalize to unseen smart devices.

Appendix C.8 provides more information about the dataset.

## 4.6 PedCount: Forecasting of pedestrian crossings throughout locations

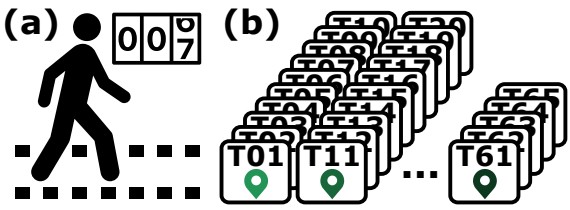

Figure 10: Summary of the PedCount dataset. (a) The task is to forecast the count of pedestrian crossing streets of Melbourne. (b) The dataset has 65 source domains, where each domain contains pedestrian counts of a different street crossing. The goal is to perform well on unseen street crossings.

Data gathered from the behavior of a population follows seasonal (daily, weekly, yearly) trends. An example of this is the movement of population within a city, either by walking, public transport or car. These trends form from the daily life of the population, e.g., the influx in the morning, outflux in the evening, and absence on the weekend. However, these trends can shift when the data is gathered from different sources in a city. We study the impact of those trend shifts with the PedCount (City of Melbourne, 2017; Godahewa et al., 2021) dataset (Figure 10).

The dataset has 65 source domains, where each domain contains pedestrian counts of a different street crossing. The goal is to perform well on unseen street crossings. Specifically, we investigate the OOD generalization to location T22 and T25.

Appendix C.9 provides more information about the dataset.

## 4.7 AusElec: Forecasting of energy consumption throughout the year

Seasonality is the property of time series where recurring characteristics appear every cycle of a fixed period, e.g., weekly. A common practice in the forecasting field is to provide models with additional information, e.g., day of week in order to allow models to leverage seasonality for better predictions. However, holidays is a seasonality of time series that is very sparse which models often fail to capture. We study the performance of

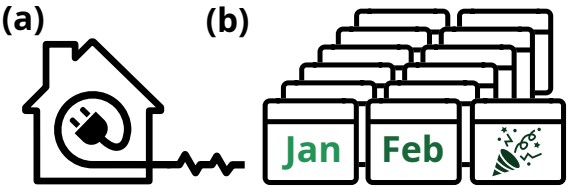

Figure 11: Summary of the AusElec dataset. (a) The task is to forecast electricity consumption. (b) The dataset has 13 time domains, where each domain contains data from different months and holidays. The goal is to perform well on all seasonalities.

models on sparse seasonality with the AusElec (Hyndman & Athanasopoulos, 2018; Godahewa et al., 2021) dataset (Figure 11)

We consider the electricity consumption forecasting task. The dataset has 13 time domains, where each domain contains data from different months and holidays. The goal is to perform well on all seasonalities.

Appendix C.10 provides more information about the dataset.

### 4.8 IEMOCAP: Emotion recognition across different conversational emotion shifts

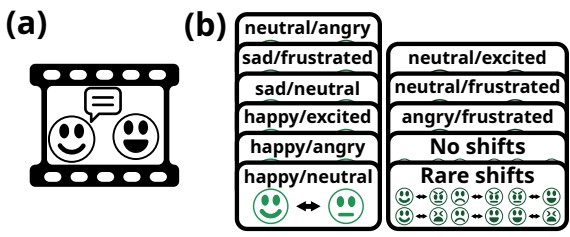

Figure 12: Summary of the IEMOCAP dataset. (a) The task is to perform emotion recognition from multi modal data (video, sound, text). (b) The dataset has 11 time domains, where each domain contains data from a different emotion shifts during conversations. The goal is to perform well on all conversational emotion shifts.

Speakers tend to maintain an emotional state over a conversation. However, external stimuli can invoke a shift in the emotional state of speakers (Poria et al., 2019). Such emotion shift are often sparsely represented in the data, making it hard for models to classify them adequately. Recent work on emotion recognition models (Poria et al., 2019; 2018; Majumder et al., 2019) show the failure of existing models to adapt to those emotion shift. We study the performance of models on emotional shift with the IEMOCAP (Bulut et al., 2008) dataset (Figure 12).

We consider the emotion recognition task. The dataset has 11 time domains, where each domain contains data from a different emotion shift during conversations. The goal is to perform well on all conversational emotion shifts.

Appendix C.11 provides more information about the dataset.

## 5 Adaptation of OOD generalization algorithms to time series

Many algorithms were proposed to address the failure of machine learning models under distribution shifts. However, they were formulated for the image domain and require adaptation to be used with time series. We now describe how we adapt them to the time series settings.

On top of Empirical Risk Minimization (**ERM**, Vapnik (1998)), we have selected commonly used algorithms from the OOD generalization research field to adapt and evaluate on WOODS benchmarks: Invariant Risk Minimization (**IRM**, Arjovsky et al. (2020)), Group Distributionally Robust Optimization (**GroupDRO**, Sagawa et al. (2020)), Variance Risk Extrapolation (**VREx**, Krueger et al. (2021)), Spectral Decoupling (**SD**, Pezeshki et al. (2021)), Information Bottleneck Empirical Risk Minimization (**IB-ERM**, Ahuja et al. (2021)), Transfer (**Transfer**, Zhang et al. (2021a)), Contrastive Adversarial Domain bottleneck (**CAD**, Ruan et al. (2021)), Conditional CAD (**CondCAD**, Ruan et al. (2021)), Conditional Contrastive Domain Generalization (**CCDG**, Ragab et al. (2022)), Diversify (**Diversify**, Lu et al. (2023)).

The loss function of above algorithms (except GroupDRO and Transfer) comprises of two terms: the empirical risk for a domain $R^d(f)$ and a penalty function $P(f)$. For the empirical risk of domain $d$, we average the risk

across the set of labeled time steps of a time series belonging to domain $d$: $S_p^d$.

$$R^d(f) = \frac{1}{n^d} \sum_{(\mathbf{X},\mathbf{Y}) \in D} \frac{1}{|S_p^d|} \sum_{t \in S_p^d} \mathcal{L}\big(f(X_{1:t}), Y_t\big) \tag{1}$$

where $n^d$ is the number of samples from domain $d$ in the dataset $D$. In the case of Source-domains, all time steps of a time series belongs to the same domain, while for the Time-domains there can be time steps belonging to different domains in the time series. IRM and VREx use a penalty that relies on the risk across domains, we use the risk from Equation (1) in the corresponding penalties.

$$P(f) = \frac{1}{n^d} \sum_{(\mathbf{X},\mathbf{Y}) \in D} \frac{1}{|S_p^d|} \sum_{t \in S_p^d} \tilde{P}(f, X_{1:t}, Y_t), \tag{2}$$

where $\tilde{P}$ is the penalty applied at each prediction point, e.g., $\tilde{P}(f, X_{1:t}, Y) = \|f(X_{1:t})\|^2$ for SD. Equations (1) and (2) are a simplifications of the adaptation; in Appendix D we provide a more general formulation along with explicit penalty definitions for all algorithms used in this work.

## 6 Experiments

Our framework follows the DomainBed (Gulrajani & Lopez-Paz, 2020) workflow for hyperparameter search and model selection for a fair and systematic evaluation of OOD generalization algorithms. We perform a random search over 20 hyperparameter configurations, which we repeat three times for error estimation. We then report the performance of the model chosen with our model selection methods (see Section 6.1). Table 2 summarizes the technical characteristics and backbone used for every datasets in our experimentation (See Appendix C for full details of each dataset.)

Appendix F provides more information on the on the framework along with hyperparameter search spaces.

Table 2: Technical characteristic summary of WOODS datasets

| | Spur.-Fourier | TCM.-Source | TCM.-Time | CAP | SEDFx | PCL | LSA64 | HHAR | PedCount | AusElec | IEMOCAP |
|---|---|---|---|---|---|---|---|---|---|---|---|
| Task | Classification | Classification | Classification | Classification | Classification | Classification | Classification | Classification | Forecasting | Forecasting | Classification |
| Num. Samples | 12,000 | 17,500 | 17,500 | 40,390 | 238,712 | 22,598 | 3,200 | 13,674 | - | - | 7,433 |
| Num. classes | 2 | 2 | 2 | 6 | 6 | 2 | 64 | 6 | - | - | 6 |
| Domain type | Source | Source | Time | Source | Source | Source | Source | Source | Source | Source | Time |
| Num. domains | 3 | 3 | 3 | 5 | 4 | 3 | 5 | 5 | 65 | 13 | 11 |
| Sequence length | 50 | 4 | 4 | 3000@100Hz | 3000@100Hz | 752@250Hz | 20 | 500@Hz | $500@\frac{1}{hour}$ | $500@\frac{1}{30min}$ | Varies |
| Steps shape | (1) | (2,28,28) | (2,28,28) | (19) | (4) | (48) | (3,224,224) | (6) | (1+45) | (1+47) | (712) |
| Prediction times | End | [2nd,3rd,4th] | [2nd,3rd,4th] | End | End | End | End | End | 24 steps ahead | 48 steps ahead | All steps |
| Backbone | LSTM | CNN+LSTM | CNN+LSTM | CNN | CNN | CNN | CNN+LSTM | CNN | Transformer | Transformer | Multimodal LSTM |
| Further details | Appx C.1 | Appx C.2 | Appx C.3 | Appx C.4 | Appx C.5 | Appx C.6 | Appx C.7 | Appx C.8 | Appx C.9 | Appx C.10 | Appx C.11 |

### 6.1 Model selection methods

**For domain generalization** We split all the training domains into training and validation sets. With *train-domain validation*, we choose the model that gets the best average validation performance across training domains. With *test-domain validation*, we choose the model with the best performance on the test domain, however, we restrict the test domain queries to the final training checkpoint only, effectively disallowing early stopping. With *oracle train-domain validation*, we choose the model with the best performance on the test domain, however, we restrict the test domain queries to the training checkpoint with the best performance on the validation set of the training domains.

**For subpopulation shift** We split all domains into training, validation and test sets. With *domain-average validation*, we choose the model with the best average validation performance across domains. With *worst-domain validation*, we choose the model with the best worst domain performance.

Appendix G provides more details on the model selection methods, and why we chose them.

Table 3: Summary of baseline algorithms performance on the real-world domain generalization datasets.

| | Train-domain validation | | | | | | | Oracle train-domain validation | | | | | |
|---|---|---|---|---|---|---|---|---|---|---|---|---|---|
| Objective | CAP (accuracy) | SEDFx (accuracy) | PCL (accuracy) | LSA64 (accuracy) | HHAR (accuracy) | Average | Objective | CAP (accuracy) | SEDFx (accuracy) | PCL (accuracy) | LSA64 (accuracy) | HHAR (accuracy) | Average |
| ERM | 62.8 (0.6) | 67.3 (0.8) | 64.3 (0.5) | 53.4 (2.0) | 84.4 (0.6) | 66.4 | ERM | 64.2 (0.6) | 68.5 (0.3) | **65.3** (0.3) | 58.2 (0.9) | 85.3 (0.5) | 68.3 |
| IRM | 58.7 (1.3) | 62.7 (0.7) | 63.9 (0.2) | 45.0 (1.6) | 82.9 (0.9) | 62.6 | IRM | 60.5 (0.9) | 64.3 (0.6) | 64.4 (0.4) | 43.6 (2.0) | 83.4 (0.6) | 63.2 |
| VREx | 48.6 (1.7) | 56.1 (1.4) | 63.2 (0.3) | 46.8 (2.9) | 83.2 (0.5) | 59.6 | VREx | 49.3 (1.6) | 57.0 (0.7) | 63.3 (0.3) | 50.0 (0.8) | 83.2 (0.6) | 60.6 |
| GroupDRO | 62.0 (0.8) | 65.2 (0.8) | **64.8** (0.3) | 46.3 (2.1) | 84.2 (0.4) | 64.5 | GroupDRO | 62.9 (0.6) | 66.1 (0.5) | 64.5 (0.3) | 54.0 (1.3) | 84.3 (0.4) | 66.4 |
| IB-ERM | **63.2** (0.8) | 69.5 (0.5) | 64.4 (0.3) | **57.3** (1.9) | 83.5 (0.7) | **67.6** | IB-ERM | 65.2 (0.6) | 70.6 (0.4) | 65.0 (0.3) | **59.8** (1.0) | 85.5 (0.2) | **69.2** |
| SD | 60.8 (0.9) | 69.8 (0.5) | 64.4 (0.2) | 50.7 (1.7) | **85.6** (0.1) | 66.2 | SD | 63.2 (0.4) | 70.6 (0.4) | **65.3** (0.12) | 58.6 (1.0) | 86.3 (0.2) | 68.8 |
| CAD | 62.2 (1.1) | 66.1 (0.5) | 64.6 (0.6) | 50.3 (2.2) | 85.0 (0.6) | 65.6 | CAD | 63.6 (0.8) | 67.5 (0.3) | 64.5 (0.4) | 57.8 (1.5) | 84.8 (0.3) | 67.7 |
| CondCAD | 62.6 (0.5) | 66.1 (0.8) | 64.2 (0.3) | 53.4 (1.5) | 84.3 (0.8) | 66.1 | CondCAD | 63.3 (0.7) | 66.6 (0.6) | 63.6 (0.3) | 57.4 (1.3) | 84.7 (0.6) | 67.1 |
| Transfer | 55.0 (1.3) | 61.0 (0.9) | 62.3 (0.2) | 47.3 (1.3) | 84.4 (0.5) | 62.0 | Transfer | 57.7 (0.8) | 61.5 (0.7) | 61.9 (0.2) | 51.3 (1.6) | 85.1 (0.3) | 63.5 |
| CCDG | 61.7 (1.0) | 68.2 (0.6) | 64.3 (0.2) | 53.0 (1.2) | 84.7 (0.7) | 66.4 | CCDG | 63.1 (0.6) | 69.2 (0.4) | 64.4 (0.5) | 56.0 (1.6) | 85.8 (0.3) | 67.7 |
| Diversify | 57.4 (1.9) | **76.9** (0.1) | 64.4 (0.4) | 48.6 (1.8) | 85.2 (0.7) | 66.5 | Diversify | 62.3 (1.1) | **77.2** (0.1) | 64.2 (0.4) | 50.6 (1.3) | **86.7** (0.6) | 68.2 |

Table 4: Summary of baseline algorithms performance on the synthetic challenge domain generalization datasets.

| | Train-domain validation | | | | | Test-domain validation | | | | |
|---|---|---|---|---|---|---|---|---|---|---|
| Objective | Spur.-Fourier (accuracy) | TCM.-Source (accuracy) | TCM.-Time (accuracy) | Average | Objective | Spur.-Fourier (accuracy) | TCM.-Source (accuracy) | TCM.-Time (accuracy) | Average |
| ERM | 9.9 (0.1) | 10.1 (0.0) | 9.9 (0.1) | 10.0 | ERM | 12.1 (2.0) | 30.3 (0.8) | 28.6 (2.4) | 23.7 |
| IRM | 10.3 (0.1) | 9.8 (0.1) | 10.3 (0.1) | 10.1 | IRM | 58.8 (2.0) | **52.7** (0.6) | **50.6** (0.2) | 54.1 |
| VREx | 10.4 (0.2) | 9.8 (0.2) | 9.8 (0.1) | 10.0 | VREx | 63.7 (0.7) | 49.7 (0.2) | **50.6** (0.6) | **54.7** |
| GroupDRO | 10.1 (0.2) | 10.4 (0.0) | 10.1 (0.1) | 10.2 | GroupDRO | 21.5 (2.1) | 33.5 (2.9) | 24.8 (3.9) | 26.6 |
| IB-ERM | 9.2 (0.3) | 10.0 (0.1) | 10.2 (0.0) | 9.8 | IB-ERM | 18.6 (4.0) | 28.1 (1.1) | 33.7 (6.5) | 26.8 |
| SD | 9.7 (0.2) | 10.2 (0.1) | 10.2 (0.1) | 10.0 | SD | 10.0 (0.1) | 27.4 (3.5) | 31.8 (5.1) | 23.0 |
| CAD | 10.3 (0.4) | 9.8 (0.1) | 9.9 (0.1) | 10.0 | CAD | 20.4 (2.4) | 26.3 (3.3) | 27.2 (2.3) | 28.0 |
| CondCAD | 10.3 (0.6) | 10.1 (0.1) | 9.9 (0.2) | 10.1 | CondCAD | 16.0 (1.6) | 20.6 (2.5) | 22.7 (2.9) | 20.1 |
| Transfer | 9.5 (0.1) | 10.0 (0.2) | 9.8 (0.0) | 9.8 | Transfer | 13.4 (2.8) | 18.3 (0.8) | 24.2 (5.3) | 21.6 |
| CCDG | 11.2 (0.7) | 10.1 (0.0) | 9.9 (0.1) | 10.4 | CCDG | 50.6 (0.2) | 49.8 (0.3) | 49.2 (0.3) | 49.9 |
| Diversify | **12.2** (0.9) | 9.9 (0.1) | 10.2 (0.0) | 10.7 | Diversify | **67.0** (3.2) | 29.1 (1.4) | 29.8 (2.1) | 42.0 |

## 6.2 OOD generalization algorithms results

**WOODS datasets have a significant generalization gap** Table 1 summarizes the generalization gap for all WOODS datasets, along with the In-Distribution (ID) and OOD performance. We compute the generalization gap to be an upper bound of the attainable performance on the test domains. This is positively indicative that there is significant improvements to be made over ERM.

Appendix E provide more details on how the generalization gaps are obtained.

**Marginal improvement over ERM on WOODS real-world datasets** Table 3, 5 and 6 summarizes the baseline results on our real-world datasets[1]. We observe a marginal improvement over ERM on several datasets with the adapted algorithms.

---

[1] Performance of SD, IRM, CAD, CondCAD, and Transfer are not reported on forecasting datasets, because their adaptation to a forecasting task is not possible without significant alterations to their formulations.

Table 5: Summary of baseline algorithms performance on subpopulation shifts datasets.

| | Domain-average validation | | | Worst-domain validation | | |
|---|---|---|---|---|---|---|
| Objective | AusElec (rmse) | IEMOCAP (accuracy) | Objective | AusElec (rmse) | IEMOCAP (accuracy) | |
| ERM | 397 (9) | 57.7 (1.9) | ERM | 404 (7) | 56.3 (2.8) | |
| IRM | X | 55.9 (1.2) | IRM | X | **58.9** (1.1) | |
| VREx | 415 (10) | 59.4 (1.4) | VREx | 409 (4) | 57.7 (3.1) | |
| GroupDRO | 409 (2) | 56.1 (1.2) | GroupDRO | 424 (13) | 58.8 (1.0) | |
| IB-ERM | **394** (2) | **59.9** (0.5) | IB-ERM | **391** (5) | 58.8 (1.5) | |
| SD | X | 58.0 (0.4) | SD | X | 56.1 (1.2) | |

Table 6: Summary of baseline algorithms performance on forecasting domain generalization datasets.

| Train-domain validation | | Oracle train-domain val. | |
|---|---|---|---|
| Objective | PedCount (rmse) | Objective | PedCount (rmse) |
| ERM | 204.1 (11.4) | ERM | 223.2 (7.1) |
| VREx | **201.6** (6.0) | VREx | 213.1 (3.1) |
| GroupDRO | 243.2 (13.0) | GroupDRO | 242.1 (9.9) |
| IB-ERM | 213.1 (10.9) | IB-ERM | **205.7** (11.3) |

**Algorithms fail on synthetic challenge dataset with train-domain validation**    Table 4 summarizes the baseline results on our synthetic challenge datasets. We observe that IRM and VREx significantly outperform ERM on WOODS synthetic datasets with test-domain validation. However, all algorithms fail with train-domain validation as chosen models learned to rely on the spurious features which were anti-correlated with the label during testing. This caused accuracies of 10%, significantly below the random guessing accuracy of 50%. We also observe that IRM and VREx under perform on real-world dataset.

### 6.3   Discussion and future research directions

Recent advancements in the fields of computer vision (CV) and natural language processing (NLP) have witnessed remarkable performance gains through the utilization of large-scale web data for training models (Brown et al., 2020a; Rae et al., 2021; Chowdhery et al., 2022; Radford et al., 2021b; Wei et al., 2021). This approach, which involves training models on diverse and abundant data (Kaplan et al., 2020), has demonstrated enhanced OOD generalization capabilities across various tasks and domains (Miller et al., 2021). However, when it comes to time series data, the transferability of concepts learned from self-supervision is not guaranteed (Ma et al., 2023a), posing challenges for achieving similar scaling benefits in time series OOD generalization. As a result, it remains uncertain whether a solution analogous to CV or NLP exists for effectively addressing OOD generalization in the context of time series data. In the scenario where research advances are able to achieve universal pre-training in time series, we hope WOODS can be a reliable evaluation ground for the development of these foundation models.

In the scenario that foundation models in time series remain unattainable for the near future, opportunities for alternative research directions aimed at tackling the OOD generalization problem arise. One promising avenue for exploration involves the deliberate construction of pre-training datasets tailored specifically for downstream tasks with distribution shift (Kostas et al., 2021; Malkiel et al., 2022). By designing datasets that align closely with the characteristics of the target tasks, it could be possible to maximize the transfer of learned representations to tackle distribution shift. This approach offers a potential proxy for achieving the benefits of large-scale pre-training. Consequently, investigating the construction and utilization of task-specific pre-training datasets represents an interesting direction for further research in this area.

Moreover, gaining a deeper understanding of the distribution shifts that occur in time series OOD generalization is essential for advancing the field. Similar studies conducted in the domain of computer vision (Ye et al., 2021b; Ruan et al., 2021; Ahuja et al., 2020b) have yielded valuable insights into the characteristics of distribution shift. Building upon this knowledge and exploring how these insights can be translated and applied to the unique challenges and characteristics of time series data could unlock more effective methodologies and algorithms.

Several factors contribute to the complexity of establishing a universal representation for time series data. Characteristics such as seasonality, frequencies, sampling rate, signal amplitude, and dimensionality introduce inherent challenges that impact the generalizability of models. However, maybe such factors could be leveraged with expert knowledge as part of the solution. By incorporating domain-specific insights and inductive biases derived from these characteristics, significant improvements in OOD generalization performance might be possible.

Furthermore, investigating the influence of model architecture on OOD generalization is another key aspect that warrants thorough analysis. Understanding the relationship between the architectural choices of models and their ability to generalize OOD can provide valuable insights for addressing the challenges of OOD generalization in time series. This knowledge could guide the development of novel architectures or modification strategies that are specifically tailored to enhance OOD generalization capabilities in time series data.

## 7   Conclusion & Limitations

This work introduced WOODS: a benchmark of 11 datasets for OOD generalization in time series. We formulated the Source- and Time-domain settings for dealing with different scenarios of distribution shifts in time series. We adapted OOD algorithms to the time series setting, and provided their performance on

WOODS datasets using our fair and systematic evaluation framework. With WOODS, we take the first step and lay the groundwork towards understanding and solving distribution shifts failure mode of deep learning in time series.

While this work proposes an initial set of benchmarks for OOD generalization in time series, our benchmarks are inherently biased toward Source-domains problems, classification tasks, and neurophysiology modalities. We hope for WOODS to be a platform to continue building towards a complete set of benchmarks with datasets covering those missing settings and other data modalities not currently studied in WOODS.

**Broader Impact Statement**

Failures of deep learning models under distribution shifts are very concerning in real-life applications that directly impact human lives, such as medicine or self-driving cars. The WOODS benchmark hopes to give researchers and engineers a meaningful measure of generalization performance to test new algorithms and alleviate potentially dangerous failures. However, it is possible that our benchmark does not accurately reflect OOD generalization performance for all possible applications. This could lead to false confidence in a deployed system that could be dangerous to human life.

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

## Ethical concern address

**Comments on personally identifiable information or sensitive personally identifiable information**

- **EEG datasets** (CAP, SEDFx and PCL) The dataset does not hold personally identifiable information.

- **HHAR**: The dataset does not hold personally identifiable information.

- **LSA64** The dataset holds videos of signers, which might be considered as personally identifiable information. However, because of the naturs of the data gathering process, i.e. deliberate head shot of the signers and that the dataset is openly distributed by the author at `http://facundoq.github.io/datasets/lsa64/` we can assume that participants gave permission to distribute their videos. Efforts were made to speak with Ronchetti et al. (2016) on the subject, but we were not able to make contact.

- **AusElec** The dataset does not hold personally identifiable information.

- **Pedestrian** The dataset does not hold personally identifiable information.

- **IEMOCAP** The dataset holds videos and voice recording of actors, which might be considered as personally identifiable information. However, it is reasonable to assume that actors gave consent to the data gathering because of the nature of the data, i.e., acted dialogues by actors. In any case, the data is protected by a release and will not be distributed by us as to protect property and identity of actors.

**Comments on consent to use or share the data**

- **EEG datasets** (CAP, SEDFx and PCL) the datasets were accessed through open forums of dataset (Physionet (Goldberger et al., 2000) for CAP and SEDFx, and MOABB (Jayaram & Barachant, 2018) for PCL) and thus we can assume that we have consent to use. Additionally, licenses allows us to use and distribute derived products of the data.

- **HHAR** was accessed through the UCI (Dua & Graff, 2017) open forums and thus we assume that we have consent to use.

- **LSA64** is openly distributed by the author at `http://facundoq.github.io/datasets/lsa64/` which allows us to assume that we have consent to use. The dataset has a *Attribution-NonCommercial-ShareAlike 4.0 International* license which gives us permission to distribute derived products. Efforts were made to speak with Ronchetti et al. (2016) to confirm assumptions on the subject, but we were not able to make contact.

- **AusElec** was accessed through the Monash time series archive (Godahewa et al., 2021). We obtained direct consent from Godahewa et al. (2021) to use the dataset in our work.

- **Pedestrian** was accessed through the Monash time series archive (Godahewa et al., 2021). We obtained direct consent from Godahewa et al. (2021) to use the dataset in our work.

- **IEMOCAP**: We obtained direct consent to use under the license agreement on their website (`https://sail.usc.edu/iemocap/`). However, we will not ourselves be distributing this dataset, users will need to sign the release form and obtain the dataset themselves from SAIL at USC (`https://sail.usc.edu/`) in order to use it in WOODS. This is to protect property and identity of actors.

# A   Organization

In Appendix B, we provide additional related works, including works in both OOD generalization algorithms and existing datasets in the field. In Appendix C, we provide further details on all WOODS datasets, along with model architecture choices and licenses. In Appendix D, we provide a general formulation for OOD generalization algorithms adaptation to time series, along with explicit penalty value function definitions for the algorithms used in this work. In Appendix E, we give further details on how we define the generalization gaps in our datasets. In Appendix F, we describe our evaluation framework. In Appendix G, we discuss the model selection strategies used in this work.

# B   Related works

In the main text, we covered important benchmarks in the field of OOD generalization. In this section, we detail a broader horizon of datasets in the field along with OOD generalization algorithms.

## B.1   OOD generalization algorithms

Several algorithms were recently proposed to address the OOD generalization failures of deep learning (Arjovsky et al., 2020; Krueger et al., 2021; Pezeshki et al., 2021; Ahuja et al., 2021; Sagawa et al., 2020; Parascandolo et al., 2020; Shahtalebi et al., 2021; Koyama & Yamaguchi, 2020; Robey et al., 2021; Ruan et al., 2021; Rame et al., 2021; Ahuja et al., 2020a; Xu & Jaakkola, 2021; Müller et al., 2021; Liu et al., 2021; Lu et al., 2021; Rosenfeld et al., 2022; Chen et al., 2022; Sharifi-Noghabi et al., 2021; Ragab et al., 2022). Several of these algorithms adopt the invariance principle from causality (Pearl, 2009; 1995; Peters et al., 2016) to create predictors that rely on the causes of the label to make predictions. Invariance is leveraged because it is a more flexible and scalable alternative to conditional independence testing typically used for causal discovery (Zhang et al., 2012; Strobl et al., 2019). An optimal predictor that relies on the cause will be min-max optimal (Ahuja et al., 2020b; Müller et al., 2021; Rojas-Carulla et al., 2018) under a large class of distribution shifts. Some works have also been proposed to address the distribution shift that arises through time in time series forecasting tasks (Du et al., 2021; Wu et al., 2021; Ye & Dai, 2022). Other works look at representation learning for time series OOD generalization (Lu et al., 2023; Ma et al., 2023b).

## B.2   Existing benchmarks for OOD generalization

**Synthetic datasets**   Many synthetic and semi-synthetic datasets were created to gain a better understanding of generalization failure in deep learning, e.g., CMNIST (Arjovsky et al., 2020) investigates our motivating cow or camel classification problem, RMNIST (Ghifary et al., 2015) investigates invariance with respect to rotation of images, and Invariance Unit Tests (Aubin et al., 2021) investigates six different types of distribution shifts for linear models.

**Image datasets**   Many real (i.e., non-synthetic) image datasets were proposed, some with naturally occurring distribution shifts and some with artificially induced distribution shifts. Several of these datasets are composed of different renditions of the same underlying labels, e.g., PACS (Li et al., 2017) (Photo, Art, Cartoon, Sketch), DomainNet (Peng et al., 2019) (Clipart, Infographic, Painting, Quickdraw, Photo, Sketch), Office-Home (Venkateswara et al., 2017) (Art, Clipart, Product, Photo), and ImageNet-R (Hendrycks et al., 2021a) (art, cartoons, graffiti, embroidery). Others focus on the generalization across different datasets with same rendition, such as many altered versions of ImageNet, e.g., ImageNet-A (Hendrycks et al., 2021b) comprises of ImageNet images that are missclassified by ResNet models, ImageNet-C (Hendrycks & Dietterich, 2019) comprises algorithmically corrupted images from the original ImageNet, ImageNet-Sketch (Wang et al., 2019) comprises samples through Google Image queries, ImageNet-V2 (Recht et al., 2019) comprises similar images to ImageNet collected by closely following the original labeling protocol, BREEDS (Santurkar et al., 2020) comprises of ImageNet subclasses that are held out during training. Others created datasets of similar renditions but different sources, e.g., VLCS (Torralba & Efros, 2011) comprises images from four different photo datasets, ObjectNet (Barbu et al., 2019) comprises images from different predefined viewpoints, Terra Incognita (Beery et al., 2018) comprises images from multiple different traps. Another dataset class has

strong spurious features that create shortcuts to minimize the empirical risk, e.g., in CelebA (Liu et al., 2015) hair color as a spurious attribute to a gender classification task, while in NICO (He et al., 2019), Waterbirds (Sagawa et al., 2020) and backgrounds challenge (Xiao et al., 2020) use the background as a spurious attribute of animal classification task. Finally, some other datasets were created to study specific problems, e.g., Shift15m (Kimura et al., 2021) that looks at OOD generalization in the large data regime.

**Language datasets**  Natural language is prone to distribution shifts because of interindividual variability, consequently, many works investigated OOD generalization in language. The Machine Translation dataset from the work of Malinin et al. (2021) investigates generalization to atypical language usage in a translation task. Csordás et al. (2021) explored the systematic generalization of transformers with five datasets, i.e., SCAN (Lake & Baroni, 2018) uses splits of different sentence lengths, CFQ (Keysers et al., 2019) uses splits of different text structures, PCFG (Hupkes et al., 2020) uses different split definitions to investigate different aspects of generalization, COGS (Kim & Linzen, 2020) uses splits that can be addressed with compositional generalization, and the Mathematics dataset (Saxton et al., 2019) uses extrapolation sets to measure generalization. Hendrycks et al. (2020) showed that pretrained transformers help OOD generalization compared to other language models. They use three sentiment analysis datasets, i.e., generalization between SST-2 (Socher et al., 2013) and IMDb (Maas et al., 2011), the Yelp Review dataset with food types as domains, the Amazon Review dataset (McAuley et al., 2015; He & McAuley, 2016) with domains composed of clothing categories. They also used three reading comprehension datasets, i.e., STS-B (Cer et al., 2017) has text of different genres (news and captions), ReCoRD (Zhang et al., 2018) has news paragraphs from different news sources (CNN and Daily Mail), and MNLI (Williams et al., 2017) has text from differently communicated interactions such as transcribed telephone and face-to-face conversations.

**Temporal datasets**  Some works looked at temporal distribution shifts in different settings. In natural language processing, Lazaridou et al. (2021) investigated the ability of language models to generalize to future utterances beyond their training period on the WMT (Barrault et al., 2019) and ArXiv (Warner, 2001) datasets. In the clinical setting, both Zhang et al. (2021b) and Guo et al. (2022) investigated shifts when data is grouped according to the year in which they were gathered: the former used in-hospital mortality records and X-rays of the lungs, while the later used patients health record in the ICU. Malinin et al. (2021) investigated temporal shifts in large amounts of weather data.

**Other modalities**  There have been efforts in studying OOD generalization on graphs, such as works from Li et al. (2022) and the OGB-MolPCBA (Koh et al., 2021) dataset adapted from the Open Graph Benchmark (Hu et al., 2020).

As mentionned in Section 1, multiple works focused on gathering and standardizing datasets for a unified measure of OOD generalization algorithm performance. Gulrajani & Lopez-Paz (2020) introduced DomainBed: a collection of seven image datasets (i.e., CMNIST, RMNIST, PACS, VLCS, Office-Home, Terra Incognita, DomainNet) for a systematic OOD performance evaluation of algorithms. Ye et al. (2021b) built on top of DomainBed and added three datasets (i.e., Camelyon17-WILDS, NICO, and CelebA), along with a measure to group the datasets according to their distribution shift. Koh et al. (2021) introduced WILDS: a benchmark of several new in-the-wild distribution shifts datasets across diverse data modalities, i.e., IWildCam2020-WILDS, Camelyon17-WILDS, RxRx1-WILDS, OGB-MolPCBA, GlobalWheat-WILDS, CivilComments-WILDS, FMoW-WILDS, PovertyMap-WILDS, Amazon-WILDS, and Py150-WILDS. WILDS was recently extended with unlabeled samples for multiple of its datasets (Sagawa et al., 2021).

## C   Additional dataset information

### C.1   Spurious-Fourier

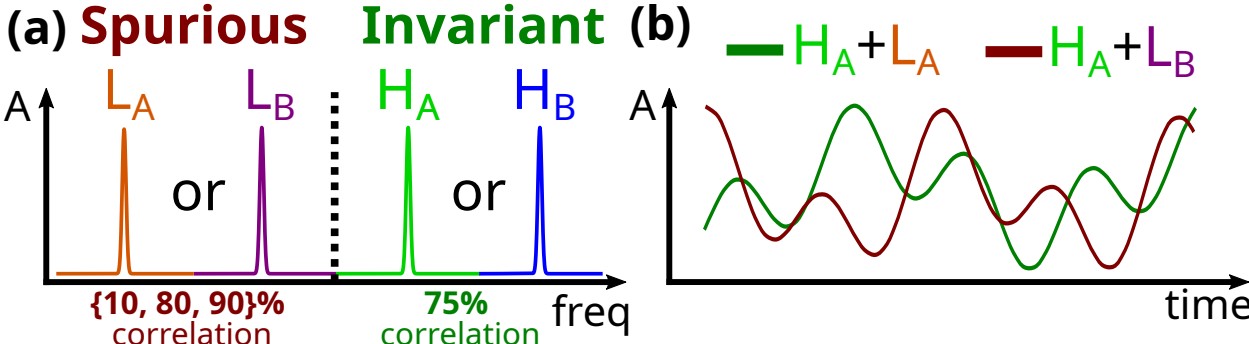

Figure 13: Description of the Spurious-Fourier dataset. Signals have one low-frequency peak and one high-frequency peak. They are then constructed from the Fourier spectrum with an inverse Fourier transform. (b) Examples of reconstructed signals, both signals have the same high frequency, but different low frequencies, which are hard to distinguish visually.

#### C.1.1   Setup

**Motivation**   Recall the cow or camel classification problem from Section 1, where a deep learning model trained to distinguish cows from camels learns to rely on the background properties (e.g., grass or sand) instead of the animal characteristic features (e.g., color) to make a prediction. Arjovsky et al. (2020) proposed Colored MNIST (CMNIST) to recreate the the cow or camel classification problem into a simple benchmark in the image domain. We propose the Spurious-Fourier dataset which is an adaptation of the cow or camel classification problem to time series.

**Problem setting**   We create a dataset composed of one-dimensional signals, where the task is to perform binary classification based on the frequency characteristics. Signals are constructed from Fourier spectra with one low-frequency peak ($L_A = 2$Hz or $L_B = 4$Hz) and one high-frequency peak ($H_A = 7$Hz or $H_B = 9$Hz), see Figure 13. Domains $D^d|_{d \in \{10\%, 80\%, 90\%\}}$ contain signal-label pairs, where the label is a noisy function of the low- and high-frequencies such that low-frequency peaks bear a varying correlation of $d$ with the label and high-frequency peaks bear an invariant correlation of 75% with the label.

**Data**   We first create four Fourier spectra with all combinations of low- and high-frequency peaks. From each of the spectra, we perform an inverse Fourier transform to get a 1 dimensional signal of 100 seconds sampled at 100Hz. We then split this long signal into smaller overlapping sequences of 50 time-steps, i.e., half a second. We then recreate the Colored MNIST (Arjovsky et al., 2020) dataset characteristic. We build datasets $D^d$ by repeating the following protocol 4000 times. First, we sample $y$ from a Bernoulli distribution $p = 0.5$. Second, we obtain $\tilde{y}$ by flipping $y$ with a probability of 25%, this gives us our high-frequency component $h$ ($\tilde{y} = 0 \rightarrow H_A$, $\tilde{y} = 1 \rightarrow H_B$). Third, we sample $z$ from a Bernoulli distribution of parameter $p = d$, this gives us our low-frequency component $l$ ($z = 0 \rightarrow L_A$, $z = 1 \rightarrow L_B$). Finally, we add to the domains dataset $D^d$ a random signal of configuration $l + h$ with the label $\tilde{y}$.

**Domain information**   Table 7 details the distribution of labels for every domain in the Spurious-Fourier dataset.

Table 7: Distribution of labels for every domain in the Spurious-Fourier dataset

| Domain | 7Hz | 9Hz | Total |
|--------|------|------|-------|
| 10% | 2043 | 1957 | 4000 |
| 80% | 2013 | 1987 | 4000 |
| 90% | 1991 | 2009 | 4000 |
| **Total** | 6047 | 5953 | 12000 |

**Architecture choice**  For this simple task, we use the LSTM (Hochreiter & Schmidhuber, 1997) model because it is a simple model well accepted in the time series/sequential prediction field. We stack on top of the LSTM a fully connected (FC) layer used to make predictions at the last time step of the time series. Layers are detailed in Table 8

Table 8: Model architecture used for the Spurious-Fourier dataset

| # | Layer |
|----|-------|
| 15 | LSTM(in=1, hidden_size=20, num_layers=2) |
| 16 | Linear(in=20, out=20) |
| 17 | ReLU |
| 18 | Linear(in=20, out=2) |

### C.1.2 Detailed results

**Oracle task**  Investigating the impact of spurious correlation in a dataset is meaningless if the underlying invariant task is impossible to solve with a given model or hyperparameter configuration. In order to avoid this, we provide the Basic-Fourier dataset in the WOODS repository. It consists of the oracle task of the Spurious-Fourier dataset, i.e., classifying 7Hz and 9Hz signals with no label noise or spurious features. We create two Fourier spectra with 7Hz and 9Hz frequency peaks respectively. From both of the spectra, we perform an inverse Fourier transform to get a one-dimensional signal of 100 seconds sampled at 100Hz. We then split this long signal into smaller overlapping sequences of 50 time-steps, i.e., half a second. While this is not a domain generalization task, the Basic-Fourier dataset is included in the WOODS repository as a sanity check that the underlying invariant task of the Spurious-Fourier dataset is possible with the model and hyperparameter configuration we are using. We show the results of ERM on the Basic-Fourier dataset in Table 9.

Table 9: Result for the Basic-Fourier dataset.

| Objective | Performance |
|-----------|-------------|
| ERM | 100.00 (0.00) |

**ID evaluation**  We evaluate the performance of ERM with access to all domains $D^d|_{d \in \{10\%, 80\%, 90\%\}}$. We obtain these results by doing a hyperparameter search with the methodology detailed in Appendix F with no held-out test domain and choose the model with train-domain validation. In other words, the training is done with all domains; thus, all domains are ID. The columns correspond to the validation accuracy of the chosen model in each domain. We see that the model learns the invariant solution to the task because the high frequencies (75%) are a stronger predictor of the label than the low frequencies (60%).

Table 10: ID results for wthe Spurious-Fourier dataset

| Algorithm | 10% | 80% | 90% | Average |
|---|---|---|---|---|
| ID ERM | 74.46 (0.07) | 74.79 (0.03) | 73.54 (0.07) | 74.26 |

**Benchmark results** We present the detailed evaluation of OOD generalization algorithms on the Spurious-Fourier dataset. Important note: We evaluate performance only when holding out the 10% domain as it is the only domain of meaning, and including the other domains only dilutes the information carried by this dataset.

Table 11: OOD generalization algorithms performance on the Spurious-Fourier dataset

| Train-domain validation | | Test-domain validation | |
|---|---|---|---|
| **Objective** | 10% | **Objective** | 10% |
| ERM | 9.91 (0.12) | ERM | 12.07 (1.99) |
| IRM | 10.30 (0.09) | IRM | 58.82 (1.98) |
| VREx | 10.36 (0.23) | VREx | 63.69 (0.70) |
| GroupDRO | 10.06 (0.19) | GroupDRO | 21.49 (2.12) |
| IB-ERM | 9.21 (0.31) | IB-ERM | 18.65 (4.01) |
| SD | 9.67 (0.20) | SD | 9.97 (0.11) |

## C.2 Temporal Colored MNIST with source domains

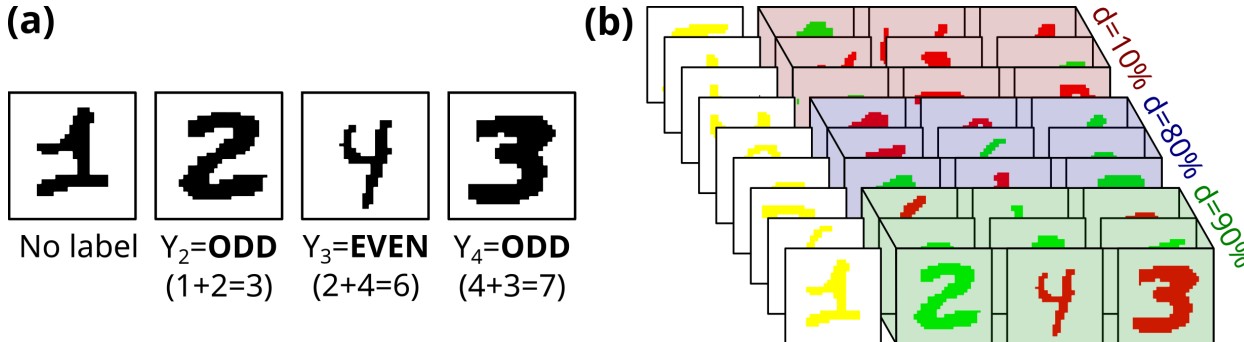

Figure 14: Description of the Temporal Colored MNIST dataset with source domains. (a) Data samples are videos of four colored MNIST digits where the task is to predict whether the sum of the current and previous digits in the sequence is odd or even. (b) Spuriously correlated color is added to each digit such that the correlation is constant among the frames of a video, but varies between video from different domains $d \in \{10\%, 80\%, 90\%\}$.

### C.2.1 Setup

**Motivation** Arjovsky et al. (2020) proposed the CMNIST dataset as a synthetic investigation of the cow or camel classification problem. We propose an extension of this widely used dataset to time series to investigate both domain definition paradigms presented in Section 2.2: Source-domains (Example 2.2) and Time-domains (Example 2.3). In this section, we give more details on the Source-domain formulation of the the dataset.

**Problem setting** In Temporal Colored MNIST with source domains (TCMNIST-Source), we create a binary classification task of video frames. Videos are sequences of four colored MNIST digits where the goal is to predict whether the sum of the current and previous digits in the sequence is odd or even, see

Figure 14(a). Prediction is made for all frames except for the first one. The label is a noisy function of the digit and color, such that the color bears a varying correlation of $d$ with the label of the frame, and the digit sums bears an invariant correlation of 75% with the label of the frame. Domains are created such that the color correlation is constant among the frames of a video, but varies between video from different domains $d \in \{10\%, 80\%, 90\%\}$. The domain definition is depicted in Figure 14(b).

**Data**   We create videos by concatenating four digits together and attributing labels $\mathbf{y}$ to the second, third and fourth frames following the parity task, see Figure 14(a). For every labeled frame $i$ in a sequence from the domain $d \in \{10\%, 80\%, 90\%\}$, we define the final label of that frame $\tilde{\mathbf{y}}_i$ by flipping the label $\mathbf{y}_i$ with a probability of 25%. Second, we define $z$ as $\tilde{\mathbf{y}}_i$ flipped with a probability equals to the domain definition (10%, 80% or 90%). Finally, we color the digit red if $z = 0$ or green if $z = 1$.

**Domain information**   Table 12 details the distribution of labels for every domain in the TCMNIST-Source dataset.

Table 12: Distribution of labels for every domain in the TCMNIST-Source dataset

| **Domain** | Even | Odd | **Domain Total** |
|---|---|---|---|
| 10% | 8603 | 8899 | 17502 |
| 80% | 8583 | 8916 | 17499 |
| 90% | 8563 | 8936 | 17499 |
| **Total** | 25749 | 26751 | 52500 |

**Architecture choice**   For this task, we use a combination of a CNN and an LSTM architecture. Table 18 details the layers of the model architecture. Its parameters were hand tuned to perform well on this task.

Table 13: Model architecture used for the TCMNIST-Source dataset

| # | Layer |
|---|---|
| 1 | Conv2D(in=d, out=8, padding=1) |
| 2 | ReLU |
| 3 | Conv2D(in=8, out=32, stride=2, padding=1) |
| 4 | ReLU |
| 5 | MaxPool2d |
| 6 | Conv2D(in=32, out=32, padding=1) |
| 7 | ReLU |
| 8 | MaxPool2d |
| 9 | Conv2D(in=32, out=32, padding=1) |
| 10 | ReLU |
| 11 | Linear(in=288, out=64) |
| 12 | ReLU |
| 13 | Linear(in=64, out=32) |
| 14 | ReLU |
| 15 | LSTM(in=32, hidden_size=128, num_layers=1) |
| 16 | Linear(in=128, out=64) |
| 17 | ReLU |
| 16 | Linear(in=64, out=64) |
| 17 | ReLU |
| 18 | Linear(in=64, out=2) |

### C.2.2 Detailed results

**Oracle task** Investigating the impact of spurious correlation in a dataset is meaningless if the underlying invariant task is impossible to solve with a given model or hyperparameter configuration. In order to avoid this, we provide the Temporal MNIST (TMNIST) dataset in the WOODS repository. It consists of the oracle task of the TCMNIST-Source dataset, i.e., classifying whether the sum of the current and last digit is odd or even without label noise and without spurious features. We create videos by concatenating four digits together and attributing labels **y** to the second, third and fourth frames following the parity task. While this is not a domain generalization task, the TMNIST dataset is included in the WOODS repository as a sanity check that the underlying invariant task of the Spurious-Fourier dataset is possible with the model and hyperparameter configuration we are using. We show the results of ERM on the TMNIST dataset in Table 14.

Table 14: Result for the TMNIST dataset

| Objective | Performance |
|-----------|-------------|
| ERM | 98.77 (0.02) |

**ID evaluation** We show the ID results of ERM for TCMNIST-Source in Table 15. We obtain these results by doing a hyperparameter search with the methodology detailed in Appendix F with no held-out test domain and choose the model with train-domain validation. In other words, the training is done with all domains; thus, all domains are ID. The columns correspond to the validation accuracy of the chosen model in each domain.

Table 15: ID results for the TCMNIST-Source dataset

| Algorithm | 10% | 80% | 90% | Average |
|-----------|-----|-----|-----|---------|
| ID ERM | 68.36 (0.13) | 73.49 (0.13) | 74.85 (0.16) | 72.23 |

**Benchmark results** We show the detailed benchmark results of the adapted OOD generalization algorithms in Table 16.

Table 16: OOD generalization algorithms performance on the TCMNIST-Source dataset

| Train-domain validation | | Test-domain validation | |
|-------------------------|--------------|------------------------|--------------|
| **Objective** | 10% | **Objective** | 10% |
| ERM | 10.07 (0.02) | ERM | 30.34 (0.82) |
| IRM | 9.82 (0.13) | IRM | 52.74 (0.59) |
| VREx | 9.83 (0.22) | VREx | 49.69 (0.25) |
| GroupDRO | 10.39 (0.02) | GroupDRO | 33.52 (2.95) |
| IB-ERM | 9.97 (0.07) | IB-ERM | 28.12 (1.12) |
| SD | 10.24 (0.12) | SD | 27.35 (3.51) |

### C.3 Temporal colored MNIST with time domains

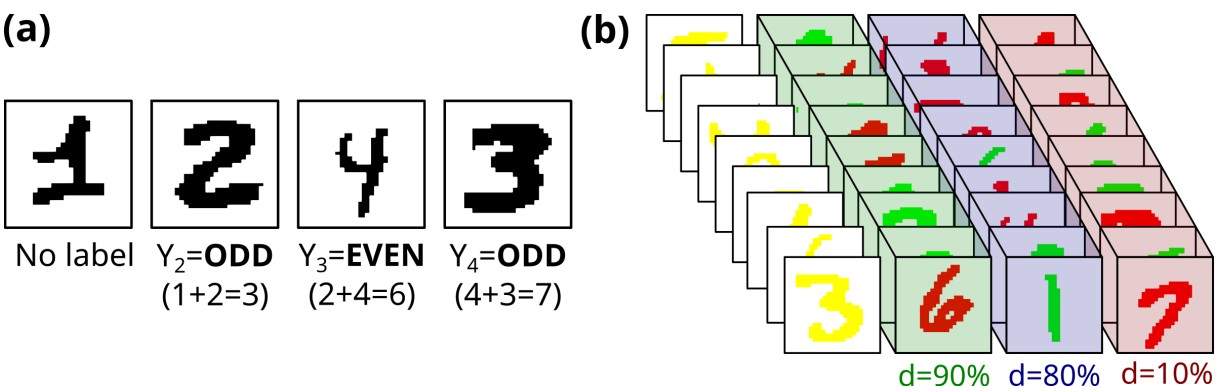

Figure 15: Description of the Temporal Colored MNIST dataset with time domains. (a) Data samples are videos of four colored MNIST digits where the task is to predict whether the sum of the current and previous digits in the sequence is odd or even. (b) Spuriously correlated color is added to each digit such that the correlation varies across frames. However, videos all have the same sequence of color correlation, where the first labeled frame correlation is 90%, second is 80% and third is 10%.

#### C.3.1 Setup

**Motivation**  Arjovsky et al. (2020) proposed the CMNIST dataset as a synthetic investigation of the cow or camel classification problem. We propose an extension of this widely used dataset to time series to investigate both domain definition paradigms presented in Section 2.2: Source-domains (Example 2.2) and Time-domains (Example 2.3). In this section, we give more details on the Time-domain formulation of the the dataset.

**Problem setting**  In Temporal Colored MNIST with time domains (TCMNIST-Time), we create a binary classification task of video frames. Videos are sequences of four colored MNIST digits where the goal is to predict whether the sum of the current and previous digits in the sequence is odd or even, see Figure 15(a). Prediction is made for all frames except for the first one. The label is a noisy function of the digit and color, such that the color bears a varying correlation of $d$ with the label of the frame, and the digit sums bears an invariant correlation of 75% with the label of the frame. Domains are created such that the color correlation varies across frames. However, videos all have the same sequence of color correlation, where the first labeled frame correlation is 90%, second is 80% and third is 10%. The domain definition is depicted in Figure 15(b).

**Data**  We create videos by concatenating four digits together and attributing labels $\mathbf{y}$ to the second, third and fourth frames following the parity task, see Figure 15(a). For every labeled frame $i \in \{2, 3, 4\}$ of all videos in the dataset, we define the final label of that frame $\tilde{\mathbf{y}}_i$ by flipping the label $\mathbf{y}_i$ with a probability of 25%. Second, we define $z$ as $\tilde{\mathbf{y}}_i$ flipped with a probability equals to the domain definition for that frame index ($i = 2 \to 90\%$, $i = 3 \to 80\%$ or $i = 4 \to 10\%$). Finally, we color the digit red if $z = 0$ or green if $z = 1$.

**Domain information**  Table 17 details the distribution of labels for every domain in the TCMNIST-Time dataset.

Table 17: Distribution of labels for every domain in the TCMNIST-Time dataset

| Domain | Even | Odd | Domain Total |
|--------|------|-----|--------------|
| 10%    | 8564 | 8936 | 17500 |
| 80%    | 8765 | 8735 | 17500 |
| 90%    | 8613 | 8887 | 17500 |
| **Total** | 25942 | 26558 | 52500 |

**Architecture choice** For this task, we use a combination of a CNN and an LSTM architecture. Table 18 details the layers of the model architecture. Its parameters were hand tuned to perform well on this toy task.

Table 18: Model architecture used for the TCMNIST-Time dataset

| # | Layer |
|---|-------|
| 1 | Conv2D(in=d, out=8, padding=1) |
| 2 | ReLU |
| 3 | Conv2D(in=8, out=32, stride=2, padding=1) |
| 4 | ReLU |
| 5 | MaxPool2d |
| 6 | Conv2D(in=32, out=32, padding=1) |
| 7 | ReLU |
| 8 | MaxPool2d |
| 9 | Conv2D(in=32, out=32, padding=1) |
| 10 | ReLU |
| 11 | Linear(in=288, out=64) |
| 12 | ReLU |
| 13 | Linear(in=64, out=32) |
| 14 | ReLU |
| 15 | LSTM(in=32, hidden_size=128, num_layers=1) |
| 16 | Linear(in=128, out=64) |
| 17 | ReLU |
| 16 | Linear(in=64, out=64) |
| 17 | ReLU |
| 18 | Linear(in=64, out=2) |

### C.3.2 Detailed results

**Oracle task** Investigating the impact of spurious correlation in a dataset is meaningless if the underlying invariant task is impossible to solve with a given model or hyperparameter configuration. In order to avoid this, we provide the Temporal MNIST (TMNIST) dataset in the WOODS repository. It consists of the oracle task of the TCMNIST-Time dataset, i.e., classifying whether the sum of the current and last digit is odd or even without label noise and without spurious features. We create videos by concatenating four digits together and attributing labels **y** to the second, third and fourth frames following the parity task. While this is not a domain generalization task, the TMNIST dataset is included in the WOODS repository as a sanity check that the underlying invariant task of the Spurious-Fourier dataset is possible with the model and hyperparameter configuration we are using. We show the results of ERM on the TMNIST dataset in Table 19.

Table 19: Result for the TMNIST dataset

| **Objective** | Performance |
|---------------|-------------|
| ERM | 98.77 (0.02) |

**ID evaluation** We show the ID results of ERM for TCMNIST-Time in Table 20. We obtain these results by doing a hyperparameter search with the methodology detailed in Appendix F with no held-out test domain and choose the model with train-domain validation. In other words, the training is done with all domains; thus, all domains are ID. The columns correspond to the validation accuracy of the chosen model in each domain.

Table 20: ID results for the TCMNIST-Time dataset

| Algorithm | 10% | 80% | 90% | Average |
|---|---|---|---|---|
| ID ERM | 89.97 (0.00) | 80.98 (0.02) | 91.20 (0.00) | 87.38 |

**Benchmark results**   We show the detailed benchmark results of the adapted OOD generalization algorithms in Table 21.

Table 21: OOD generalization algorithms performance on the TCMNIST-Time dataset

| Train-domain validation | | Test-domain validation | |
|---|---|---|---|
| **Objective** | 10% | **Objective** | 10% |
| ERM | 9.88 (0.11) | ERM | 28.61 (2.41) |
| GroupDRO | 10.09 (0.14) | GroupDRO | 24.85 (3.91) |
| IB-ERM | 10.19 (0.04) | IB-ERM | 33.70 (6.49) |
| IRM | 10.27 (0.05) | IRM | 50.65 (0.17) |
| SD | 10.19 (0.14) | SD | 31.76 (5.15) |
| VREx | 9.80 (0.06) | VREx | 50.57 (0.59) |

## C.4   CAP

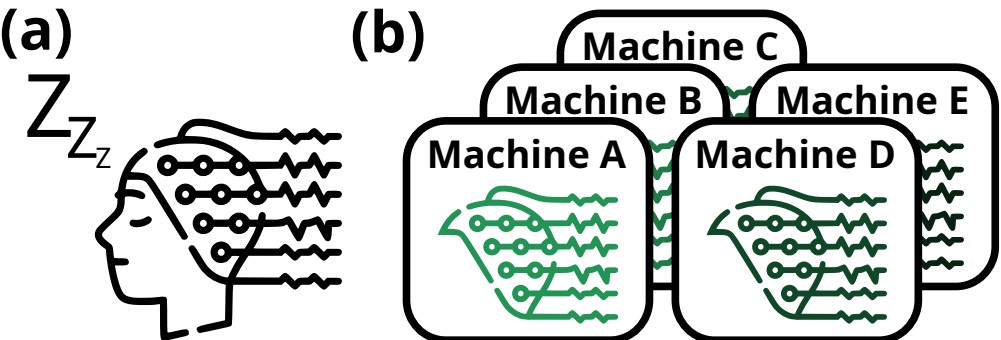

Figure 16: Summary of the CAP dataset. (a) The task is to perform sleep stage classification from EEG measurements. (b) The dataset has five source domains, where each domain contains data gathered with a different machine. The goal is to generalize to unseen machines.

### C.4.1   Setup

**Motivation**   A recurrent problem in computational medicine is that models trained on data from a given recording device will not generalize to data coming from another device, even when both devices are from a similar equipment provider. Failure to generalize to unseen machines can cause critical issues for clinical practice because a false sense of confidence in a model could lead to a false diagnosis (Kim et al., 2018; Engemann et al., 2018). We study these machinery-induced distribution shifts with the CAP (Terzano et al., 2001; Goldberger et al., 2000) dataset (Figure 5).

**Problem setting**   We consider the sleep stage classification task from electroencephalographic (EEG) measurements. The dataset has five source domains, where each domain contains data gathered with a different machine. The goal is to generalize to unseen machines.

**Data**   The dataset is composed of 40 390 gathered on 41 participants. Each participants had one night of sleep recorded. The inputs $\mathbf{X}$ are recordings of 30 seconds each with 19 channels sampled at 100Hz. The

channels include EEG but also include Electromyography (EMG), Electrocardiography (ECG), and heart rate measurements. The labels **Y** consist of 6 sleep stages: Awake, Non-REM 1, Non-REM 2, Non-REM 3, Non-REM 4, and REM. The domains $d$ are the 5 EEG machines: Machine A, Machine B, Machine C, Machine D, and Machine E.

**Preprocessing** This section details the preprocessing steps taken for the CAP dataset. The raw CAP dataset contains data from 15 machines, each with different channels and sampling frequency characteristics. We only use recordings from the five machines with the most data. Removing machines with less data allows us to retain a reasonable number (19) of shared channels between them. Next, we resample the data to a standard sampling frequency of 100Hz for all five machines. We then apply a bandpass filter from 0.3Hz to 30Hz. This bandpass filter removes frequency bands generally considered uninformative for sleep stage classification. Next, we split the nights of sleep into sequences of 30 seconds for training and testing. Finally, we then detrend and normalize the 30 second recordings with a standard scaler applied to the channels individually.

**Domain information** Table 22 details the number of participants per domain and some demographic information; each had a single night of sleep recorded. Table 23 details the proportion of samples and labels across domains.

Table 22: Number of participants and demographic information of the CAP dataset

| Domain | Number of participants | Male | Female | Age |
|---|---|---|---|---|
| Machine A | 13 | 7 | 6 | $33.1 \pm 13.9$ |
| Machine B | 5 | 5 | 0 | $26.4 \pm 8.2$ |
| Machine C | 5 | 4 | 1 | $73.4 \pm 6.42$ |
| Machine D | 10 | 5 | 5 | $30.7\,(8.9$ |
| Machine E | 8 | 3 | 5 | $36.8 \pm 16.7$ |
| **Total** | 41 | 24 | 17 | $37.3 \pm 18.1$ |

Table 23: Domain proportions of labels in the CAP dataset

| Domain | Awake | NREM 1 | NREM 2 | NREM 3 | NREM 4 | REM | **Domain Total** |
|---|---|---|---|---|---|---|---|
| Machine A | 1448 | 350 | 4986 | 1533 | 2110 | 2342 | 12769 |
| Machine B | 318 | 171 | 1933 | 595 | 706 | 971 | 4694 |
| Machine C | 1318 | 294 | 1168 | 595 | 810 | 547 | 4732 |
| Machine D | 1114 | 580 | 3547 | 1273 | 1606 | 1810 | 9930 |
| Machine E | 967 | 276 | 3377 | 711 | 1251 | 1683 | 8265 |
| **Total** | 5165 | 1671 | 15011 | 4707 | 6483 | 7353 | 40390 |

**Architecture choice** For this dataset, we use a deep convolution network model as defined in work from Schirrmeister et al. (2017). We use the implementation of the BrainDecode (Schirrmeister et al., 2017) Toolbox. We chose this model because it is the perfect combination of performance stability and recognition from the EEG community. The implementation is available at `https://github.com/TNTLFreiburg/braindecode`.

### C.4.2 Detailed results

**ID evaluation** We show the results of ERM for the CAP dataset in Table 24. We obtain these results by doing a hyperparameter search with the methodology detailed in Appendix F with no held-out test domain and choose the model with train-domain validation. In other words, the training is done with all domains;

thus, all domains are ID. The columns correspond to the validation accuracy of the chosen model in each domain.

Table 24: ID results for the CAP dataset

| Algorithm | Machine A | Machine B | Machine C | Machine D | Machine E | Average |
|---|---|---|---|---|---|---|
| ID ERM | 78.26 (0.52) | 78.14 (1.00) | 63.39 (1.38) | 78.73 (0.34) | 77.09 (0.37) | 75.12 |

**Benchmark results** We show the detailed benchmark results of the adapted OOD generalization algorithms in Table 25. Each results is obtained by holding out one domain during training and reporting the performance of the chosen model from the hyperparameter sweep on that held out domain, more details in Appendix F.

Table 25: OOD generalization algorithms performance on the CAP dataset

| Train-domain validation | | | | | | |
|---|---|---|---|---|---|---|
| **Objective** | Machine A | Machine B | Machine C | Machine D | Machine E | **Average** |
| ERM | 68.93 (0.54) | 61.98 (0.53) | 40.10 (0.75) | 73.10 (0.83) | 70.13 (0.33) | 62.85 |
| IRM | 67.59 (0.55) | 48.40 (3.14) | 41.01 (1.10) | 69.52 (1.03) | 66.86 (0.71) | 58.68 |
| VREx | 57.97 (1.92) | 38.96 (0.42) | 33.81 (1.19) | 52.53 (3.49) | 59.71 (1.53) | 48.60 |
| GroupDRO | 68.07 (0.33) | 59.22 (1.53) | 41.38 (0.52) | 72.25 (0.70) | 69.12 (0.90) | 62.01 |
| IB-ERM | 70.20 (0.71) | 62.03 (1.79) | 40.66 (0.58) | 72.73 (0.18) | 70.57 (0.83) | 63.24 |
| SD | 69.29 (0.25) | 55.53 (1.45) | 41.36 (1.78) | 71.14 (0.22) | 66.48 (0.92) | 60.76 |
| **Oracle train-domain validation** | | | | | | |
| **Objective** | Machine A | Machine B | Machine C | Machine D | Machine E | **Average** |
| ERM | 69.00 (0.51) | 65.21 (1.38) | 43.11 (0.30) | 73.31 (0.67) | 70.34 (0.16) | 64.19 |
| IRM | 67.59 (0.55) | 55.09 (2.08) | 41.20 (1.19) | 70.72 (0.46) | 67.87 (0.26) | 60.49 |
| VREx | 57.79 (2.06) | 39.49 (0.65) | 36.38 (0.57) | 52.95 (3.15) | 59.71 (1.53) | 49.26 |
| GroupDRO | 68.73 (0.22) | 60.39 (1.23) | 43.19 (0.56) | 72.51 (0.49) | 69.91 (0.50) | 62.95 |
| IB-ERM | 71.06 (0.37) | 66.99 (0.93) | 43.21 (0.76) | 73.64 (0.50) | 70.88 (0.59) | 65.16 |
| SD | 69.38 (0.20) | 61.84 (0.39) | 43.97 (1.09) | 71.41 (0.14) | 69.41 (0.29) | 63.20 |

### C.4.3 Credit and license

This dataset is adapted from the work of Terzano et al. (2001), as made available on the online Physionet (Goldberger et al., 2000) platform. This dataset is licensed under the Open Data Commons Attribution License v1.0.

### C.5 SEDFx

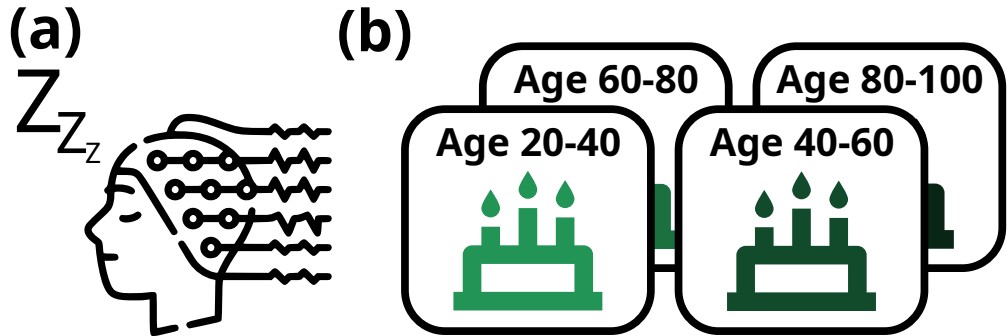

Figure 17: Summary of the SEDFx dataset. (a) The task is to perform sleep stage classification from EEG measurements. (b) The dataset has four source domains, where each domain contains data from participants of a certain age group. The goal is to generalize to unseen age groups.

#### C.5.1 Setup

**Motivation**  In clinical settings, we train a model on the data gathered from a limited number of patients and hope this model will generalize to new patients in the future (Pfohl et al., 2022). However, this generalization between observed patients in the training dataset and new patients is not guaranteed. Distribution shifts caused by shifts in patient demographics (e.g., age, gender, and ethnicity) can cause the model to fail. We study age demographic shift with the SEDFx (Kemp et al., 2000; Goldberger et al., 2000) dataset (Figure 17).

**Problem setting**  We consider the sleep classification task from EEG measurements. The dataset has four source domains, where each domain contains data from participants of a certain age group. The goal is to generalize to an unseen age demographic.

**Data**  The dataset is composed of 238 712 recordings gathered on 100 participants. Every participant had 2 nights of sleep recorded. The inputs $\mathbf{X}$ are recordings of 30 seconds each with four EEG channels sampled at 100Hz. The channels include 2 EEG channels, one Electromyography (EOG) channel, and one Electrocardiography (ECG) channel. The labels $\mathbf{Y}$ consist of 6 sleep stages: Awake, Non-REM 1, Non-REM 2, Non-REM 3, Non-REM 4, and REM. The domains $d$ are the four disjoint age groups: Age 20-40, Age 40-60, Age 60-80, and age 80-100.

**Preprocessing**  This section details the preprocessing steps taken for the SEDFx dataset. The raw SEDFx dataset contains data from 2 machines with different channels and sampling frequency characteristics. We use the data from both machines and keep only the four channels they have in common. First, we resample the data to a standard sampling frequency of 100Hz for both machines. We then apply a bandpass filter from 0.3Hz to 30Hz. This bandpass filter removes frequency bands generally considered uninformative for sleep stage classification. Next, we crop the unlabeled onset and end of the complete recordings. Next, we split the nights of sleep into shorter sequences of 30 seconds for training and testing. Finally, we detrend the data and normalize the 30 second recordings with a standard scaler applied to channels individually.

**Domain information**  The data from the different machines consists of data from disjoint sets of participants. Table 27 details the number of participants per domain and some demographic information. Table 27 details the proportion of samples and labels across domains.

Table 26: Number of participants and demographic information of the SEDFx dataset

| Domain | Number of participants | Male | Female | Age |
|---|---|---|---|---|
| Age 20-40 | 32 | 14 | 18 | $27.6 \pm 4.7$ |
| Age 40-60 | 29 | 12 | 17 | $53.3 \pm 3.4$ |
| Age 60-80 | 23 | 10 | 13 | $69.2 \pm 3.5$ |
| Age 80-100 | 16 | 8 | 8 | $90.5\,(4.5$ |
| **Total** | 100 | 44 | 56 | $54.7 \pm 22.6$ |

Table 27: Domain proportions of labels in the SEDFx dataset

| Domain | Awake | NREM 1 | NREM 2 | NREM 3 | NREM 4 | REM | Domain Total |
|---|---|---|---|---|---|---|---|
| Age 20-40 | 10505 | 4222 | 28105 | 4830 | 4254 | 12348 | 64264 |
| Age 40-60 | 20405 | 7182 | 27222 | 3243 | 1423 | 10007 | 69482 |
| Age 60-80 | 14708 | 7087 | 19186 | 2830 | 1400 | 6917 | 52128 |
| Age 80-100 | 25358 | 6684 | 14410 | 1288 | 186 | 4912 | 52838 |
| **Total** | 70976 | 25175 | 88923 | 12191 | 7263 | 34184 | 238712 |

**Architecture choice**  For this dataset, we use a deep convolution network model as defined in work from Schirrmeister et al. (2017). We use the implementation of the BrainDecode (Schirrmeister et al., 2017) Toolbox. We chose this model because it is the perfect combination of performance, stability, and recognition from the EEG community. The implementation is available at `https://github.com/TNTLFreiburg/braindecode`.

### C.5.2  Detailed results

**ID evaluation**  We show the results of ERM for the SEDFx dataset in Table 28. We obtain these results by doing a hyperparameter search with the methodology detailed in Appendix F with no held-out test domain and choose the model with train-domain validation. In other words, the training is done with all domains; thus, all domains are ID. The columns correspond to the validation accuracy of the chosen model in each domain.

Table 28: ID results for the SEDFx dataset

| Algorithm | Age 20-40 | Age 40-60 | Age 60-80 | Age 80-100 | Average |
|---|---|---|---|---|---|
| ID ERM | 74.11 (0.12) | 74.19 (0.47) | 72.39 (0.41) | 69.22 (0.49) | 72.48 |

**Benchmark results**  We show the detailed benchmark results of the adapted OOD generalization algorithms in Table 29. Each results is obtained by holding out one domain during training and reporting the performance of the chosen model from the hyperparameter sweep on that held out domain, more details in Appendix F.

### C.5.3  Credit and license

This dataset was adapted from the work of Kemp et al. (2000), as made available on the online Physionet (Goldberger et al., 2000) platform. This dataset is licensed under the Open Data Commons Attribution license v1.0.

Table 29: OOD generalization algorithms performance on the SEDFx dataset

| Train-domain validation | | | | | |
| --- | --- | --- | --- | --- | --- |
| **Objective** | Age 20-40 | Age 40-60 | Age 60-80 | Age 80-100 | **Average** |
| ERM | 65.90 (2.06) | 70.59 (0.54) | 68.48 (0.17) | 64.18 (0.33) | 67.29 |
| IRM | 60.76 (0.94) | 67.69 (0.80) | 63.04 (0.81) | 59.13 (0.43) | 62.65 |
| VREx | 57.21 (2.39) | 58.80 (0.94) | 56.81 (0.97) | 51.74 (1.23) | 56.14 |
| GroupDRO | 67.01 (0.90) | 68.31 (0.73) | 65.08 (0.42) | 60.48 (1.07) | 65.22 |
| IB-ERM | 69.41 (0.12) | 72.58 (0.46) | 69.79 (0.42) | 66.16 (0.87) | 69.48 |
| SD | 69.87 (1.34) | 73.18 (0.19) | 69.14 (0.16) | 67.18 (0.26) | 69.84 |
| **Oracle train-domain validation** | | | | | |
| **Objective** | Age 20-40 | Age 40-60 | Age 60-80 | Age 80-100 | **Average** |
| ERM | 69.87 (0.41) | 71.03 (0.20) | 68.88 (0.18) | 64.36 (0.26) | 68.53 |
| IRM | 66.02 (0.55) | 67.69 (0.80) | 63.16 (0.65) | 60.14 (0.32) | 64.25 |
| VREx | 59.63 (0.76) | 58.80 (0.94) | 56.78 (0.95) | 52.73 (0.22) | 56.99 |
| GroupDRO | 68.73 (0.37) | 69.14 (0.40) | 65.17 (0.35) | 61.54 (0.85) | 66.15 |
| IB-ERM | 70.42 (0.41) | 72.79 (0.62) | 70.25 (0.12) | 69.08 (0.56) | 70.64 |
| SD | 71.22 (0.60) | 73.18 (0.19) | 69.60 (0.04) | 68.41 (0.60) | 70.60 |

## C.6 PCL

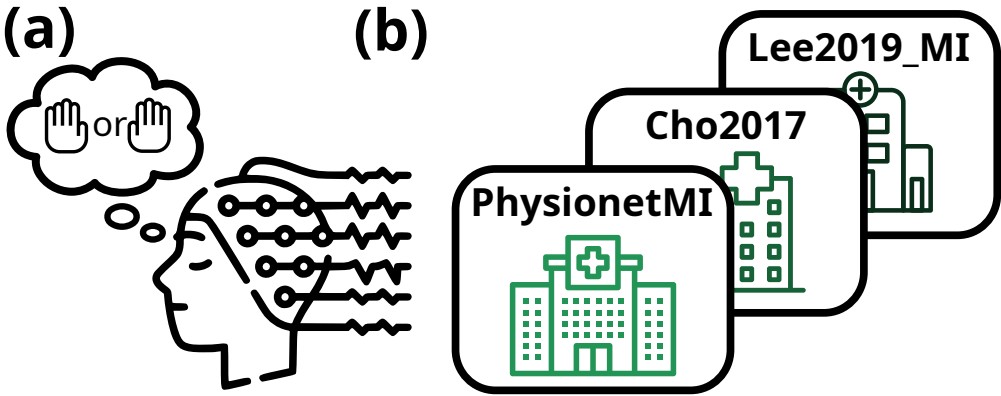

Figure 18: Summary of the PCL dataset. (a) The task is to perform motor imagery classification from EEG measurements. (b) The dataset has three source domains, where each domain contains a dataset from a different research group carrying out the same task. The goal is to generalize to unseen datasets of the same task.

### C.6.1 Setup

**Motivation** Aside from changes in the recording device and shifts in patient demographics, human intervention in the data gathering process is another contributing factor to the distribution shift that can lead to failure of clinical models (e.g., Camelyon17 (Koh et al., 2021; Sagawa et al., 2021)). This challenge is especially prevalent in temporal medical data (e.g., EEG, MEG, and others) because recording devices are complex tools greatly affected by nonlinear effects and modulations. These effects are often caused by context and preparations made before the recording (Engemann et al., 2018). We study these procedural shifts with the PCL (Lee et al., 2019; Cho et al., 2017; Schalk et al., 2004; Jayaram & Barachant, 2018) dataset (Figure 18).

**Problem setting** We consider the motor imagery task from electroencephalographic (EEG) measurements. The dataset has three source domains, where each domain contains a dataset from a different research group carrying out the same task. The goal is to generalize to unseen data gathering processes.

**Data** The dataset is composed of $22\,598$ recordings gathered with 215 participants. The inputs $\mathbf{X}$ are recordings of three seconds each with 48 EEG channels sampled at 250Hz. The 48 channels contain only EEG measurements. The labels $\mathbf{Y}$ are two imagined movements: left hand and right hand. The domains $d$ are three different motor imagery datasets: Schalk04 (Schalk et al., 2004), Cho17 (Cho et al., 2017) and Lee19 (Lee et al., 2019).

The 48 channels are: AF7, CP5, AF4, P4, P8, P2, FC6, Fz, C5, O1, Fp1, Fp2, F4, CP4, PO3, C1, FC1, T8, Pz, Oz, TP7, Cz, FC2, CP6, CP2, POz, PO4, C6, P7, AF3, FC4, TP8, CP1, O2, C2, F8, FC3, P3, AF8, FC5, F7, F3, T7, C4, CP3, CPz, C3, P1. The channel locations are shown in Figure 19.

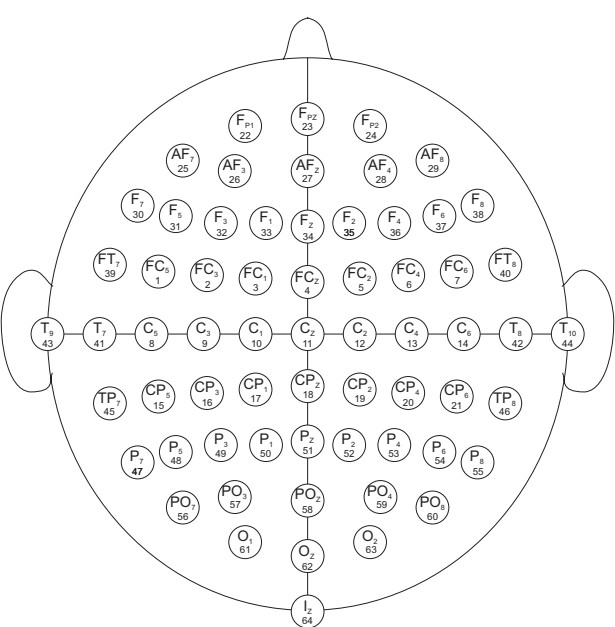

Figure 19: International 10-10 system EEG channel labeling.

**Preprocessing** This section details the preprocessing steps taken for the PCL dataset. The raw PCL dataset contains data from 2 machines, Schalk04 (Schalk et al., 2004) and Cho17 (Cho et al., 2017) both used a BCI2000 system (Schalk et al., 2004) while Lee19 (Lee et al., 2019) used an undefined machine. Both machines have different channels and sampling frequency characteristics. We take only the 48 channels they have in common and we resample the data to a standard sampling frequency of 250Hz for both machines. We then apply a bandpass filter from 0.3Hz to 30Hz. This bandpass filter removes frequency bands generally

considered uninformative for the motor imagery task. Finally, we then detrend the data and normalize the three second recordings with a standard scaler applied to the channels individually.

**Domain information**   Table 30 details the number of participants per domain and some demographic information; we put N/A for unavailable demographic information. Table 31 details the proportion of samples and labels across domains.

Table 30: Number of participants and demographic information of the PCL dataset

| **Domain** | Number of participants | Male | Female | Age |
|---|---|---|---|---|
| Schalk04 | 109 | N/A | N/A | N/A |
| Cho2017 | 52 | 33 | 19 | $24.8 \pm 3.9$ |
| Lee19 | 54 | 29 | 25 | [24, 35] |
| **Total** | 215 | N/A | N/A | N/A |

Table 31: Domain proportions of labels in the PCL dataset

| **Domain** | Left Hand | Right Hand | **Domain Total** |
|---|---|---|---|
| Schalk04 | 2480 | 2438 | 4918 |
| Cho2017 | 4940 | 4940 | 9880 |
| Lee19 | 3900 | 3900 | 7800 |
| **Total** | 11320 | 11278 | 22598 |

**Architecture choice**   For this dataset, we use a deep convolution network model as defined in work from Lawhern et al. (2018). We use the implementation of the BrainDecode Schirrmeister et al. (2017) Toolbox. We chose this model because it is well recognized by the EEG community. It also has a smaller architecture that better fits the data amount and task complexity of the PCL dataset. The implementation is available at `https://github.com/TNTLFreiburg/braindecode`.

### C.6.2   Detailed results

**ID evaluation**   We show the results of ERM for the PCL dataset in Table 32. We obtain these results by doing a hyperparameter search with the methodology detailed in Appendix F with no held-out test domain and choose the model with train-domain validation. In other words, the training is done with all domains; thus, all domains are ID. The columns correspond to the validation accuracy of the chosen model in each domain.

Table 32: ID results for the PCL dataset

| **Algorithm** | Schalk04 | Cho17 | Lee19 | **Average** |
|---|---|---|---|---|
| ID ERM | 76.40 (0.19) | 68.07 (0.09) | 76.45 (0.23) | 73.64 |

**Benchmark results**   We show the detailed benchmark results of the adapted OOD generalization algorithms in Table 33. Each results is obtained by holding out one domain during training and reporting the performance of the chosen model from the hyperparameter sweep on that held out domain, more details in Appendix F.

Table 33: OOD generalization algorithms performance on the PCL dataset

| Train-domain validation | | | | |
|---|---|---|---|---|
| **Objective** | Schalk04 | Cho17 | Lee19 | **Average** |
| ERM | 63.52 (0.92) | 59.34 (0.23) | 70.06 (0.46) | 64.31 |
| IRM | 63.43 (0.37) | 60.41 (0.07) | 67.90 (0.27) | 63.91 |
| VREx | 62.65 (0.29) | 58.84 (0.26) | 68.22 (0.33) | 63.24 |
| GroupDRO | 63.97 (0.57) | 60.24 (0.35) | 70.34 (0.02) | 64.85 |
| IB-ERM | 63.31 (0.16) | 59.82 (0.38) | 70.18 (0.41) | 64.44 |
| SD | 63.72 (0.20) | 59.31 (0.36) | 70.15 (0.16) | 64.40 |
| **Oracle train-domain validation** | | | | |
| **Objective** | Schalk04 | Cho17 | Lee19 | **Average** |
| ERM | 64.52 (0.25) | 60.41 (0.22) | 71.11 (0.29) | 65.35 |
| IRM | 63.28 (0.30) | 61.09 (0.42) | 68.77 (0.41) | 64.38 |
| VREx | 62.41 (0.47) | 59.28 (0.29) | 68.08 (0.20) | 63.26 |
| GroupDRO | 63.96 (0.36) | 59.60 (0.38) | 70.09 (0.16) | 64.55 |
| IB-ERM | 64.63 (0.22) | 60.22 (0.38) | 70.27 (0.34) | 65.04 |
| SD | 64.50 (0.05) | 60.80 (0.17) | 70.72 (0.14) | 65.34 |

### C.6.3 Credit and license

This dataset is built from 3 different motor imagery datasets (Schalk et al., 2004; Cho et al., 2017; Lee et al., 2019) as made available on the online MOABB (Jayaram & Barachant, 2018) platform. The PhysionetMI dataset is licensed under the Open Data Commons Attribution license v1.0.

### C.7 LSA64

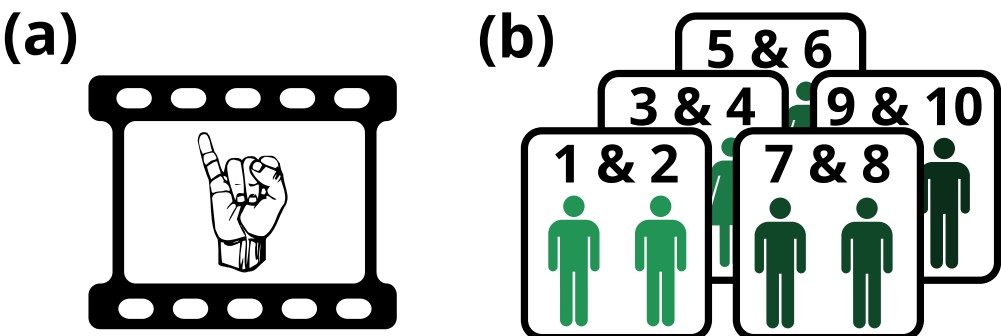

Figure 20: Summary of the LSA64 dataset. (a) The task is to perform signed word classification from videos. (b) The dataset has five source domains, where each domain contains videos of different signers. The goal is to generalize to unseen signers.

### C.7.1 Setup

**Motivation** Communication is an individualistic way to convey information through different media: text, speech, body language, and many others. However, some media are more distinctive and challenging than others. For example, text communication has less inter-individual variability than body language or speech. If deep learning systems hope to interact with humans effectively, models need to generalize to new and evolving mannerisms, accents, and other subtle variations in communication that significantly impact the

meaning of the message conveyed. We study the ability of models to recognize information coming from unseen individuals with the LSA64 (Ronchetti et al., 2016) dataset (Figure 20).

**Problem setting**   We consider the video classification of signed words in Argentinian Sign Language. The dataset has five source domains, where each domain contains videos of different signers. The goal is to generalize to unseen signers.

**Data**   The dataset consists of 3200 videos from 10 different signers signing in Argentinian Sign Language. The inputs **X** are videos of 20 frames with resolution (3, 224, 224). Sequences are two and a half seconds long. The labels **Y** consist of 64 words: Opaque, Red, Green, Yellow, Bright, Light-blue, Colors, Light-red Women, Enemy, Son, Man, Away, Drawer, Born, learn, Call, Skimmer, Bitter, Sweet milk, Milk, Water, Food, Argentina, Uruguay, Country, Last name, Where, Mock, Birthday, Breakfast, Photo, Hungry, Map, Coin, Music, Ship, None, Name, Patience, Perfume, Deaf, Trap, Rice, Barbecue, Cady, Chewing-gum, Spaghetti, Yogurt, accept, Thanks, Shut down, Appear, To land, Catch, Help, Dance, Bathe, Buy, Copy, Run, Realize, Give, and Find. The domains $d$ are 5 subgroups of signers: Signers 1 & 2, Signers 3 & 4, Signers 5 & 6, Signers 7 & 8 and Signers 9 & 10.

**Preprocessing**   This section details the preprocessing steps taken for the LSA64 dataset. The raw LSA64 dataset contains 3200 videos, each about 3 seconds long with the resolution of 1920x1080, at 60 frames per second. We first crop all videos at precisely 2.5 seconds to have videos of the same length. This cropping does not impact the information content of the video as signers pause at the end of their signed words. We then resize the frames to 224x224. Finally, we use PyTorchVideo (Fan et al., 2021) to uniformly sample 20 frames from each video in a sequence for prediction.

**Domain information**   Table 34 details the proportion of samples and labels across domains.

Table 34: Domain proportions of labels in the LSA64 dataset

| Domain | 64 words | Domain Total |
|---|---|---|
| Signer 1 & 2 | 10 videos per word | 640 |
| Signer 3 & 4 | 10 videos per word | 640 |
| Signer 5 & 6 | 10 videos per word | 640 |
| Signer 7 & 8 | 10 videos per word | 640 |
| Signer 9 & 10 | 10 videos per word | 640 |
| **Total** | 50 videos per word | 3200 |

**Architecture choice**   We use a Convolutional Recurrent Neural Network (CRNN) for this dataset. The CRNN model has 4 model blocks: Convolutional, Recurrent, attention, and prediction. First, we feed each video frame through a frozen Resnet50 model that is pretrained on Imagenet to extract relevant features. We then feed these feature vectors sequentially to an LSTM model. Finally, push the output of the LSTM model for each frame through a self-attention layer which linearly combines the LSTM output weighed by their attention scores. We the use a fully connected network to make predictions. Table 35 details the layers of the model architecture.

Table 35: Model architecture used for the LSA64 dataset

| # | Layer |
|---|---|
| 1 | Resnet50(in=3x224x224, out=2048) |
| 2 | Linear(in=2048, out=512) |
| 3 | ReLU |
| 4 | BatchNorm(num_features=512, momentum=0.01) |
| 5 | Linear(in=512, out=512) |
| 6 | ReLU |
| 7 | BatchNorm(num_features=512, momentum=0.01) |
| 8 | Linear(in=512, out=216) |
| 9 | ReLU |
| 10 | LSTM(in=256, hidden_size=128, num_layers=2) |
| 11 | SelfAttention(in=128, out=128) |
| 12 | Linear(in=128, out=64) |
| 13 | ReLU |
| 14 | Linear(in=64, out=64) |

### C.7.2 Detailed Results

**ID evaluation** We show the results of ERM for the LSA64 dataset in Table 36. We obtain these results by doing a hyperparameter search with the methodology detailed in Appendix F with no held-out test domain and choose the model with train-domain validation. In other words, the training is done with all domains; thus, all domains are ID. The columns correspond to the validation accuracy of the chosen model in each domain.

Table 36: ID results for the LSA64 dataset

| Algorithm | Signers 1 & 2 | Signers 3 & 4 | Signers 5 & 6 | Signers 7 & 8 | Signers 9 & 10 | Average |
|---|---|---|---|---|---|---|
| ID ERM | 90.10 (0.56) | 89.58 (1.13) | 80.21 (1.06) | 85.16 (1.61) | 87.76 (0.77) | 86.56 |

**Benchmarks results** We show the detailed benchmark results of the adapted OOD generalization algorithms in Table 37. Each results is obtained by holding out one domain during training and reporting the performance of the chosen model from the hyperparameter sweep on that held out domain, more details in Appendix F.

### C.7.3 Credit and license

This dataset was adapted from the work of Ronchetti et al. (2016). The LSA64 dataset is under the Creative Commons Attribution-NonCommercial-ShareAlike 4.0 International License.

Table 37: OOD generalization algorithms performance on the LSA64 dataset

| Train-domain validation | | | | | |
|---|---|---|---|---|---|
| **Objective** | **Signers 1 & 2** | **Signers 3 & 4** | **Signers 5 & 6** | **Signers 7 & 8** | **Signers 9 & 10** | **Average** |
| ERM | 48.50 (2.93) | 50.65 (2.28) | 47.53 (1.32) | 57.49 (2.49) | 62.96 (0.98) | 53.42 |
| IRM | 44.34 (0.60) | 43.16 (1.48) | 38.28 (2.01) | 46.88 (1.13) | 52.47 (2.78) | 45.03 |
| VREx | 42.19 (3.64) | 45.57 (1.67) | 42.06 (2.82) | 51.82 (1.95) | 52.21 (4.26) | 46.77 |
| GroupDRO | 43.62 (3.95) | 44.14 (1.87) | 43.29 (1.58) | 47.79 (1.70) | 52.73 (1.36) | 46.32 |
| IB-ERM | 55.66 (1.71) | 56.71 (2.16) | 49.80 (2.41) | 64.52 (0.61) | 59.70 (2.51) | 57.28 |
| SD | 48.63 (2.46) | 50.20 (1.60) | 40.89 (0.84) | 57.68 (2.54) | 56.32 (1.19) | 50.74 |

| Oracle train-domain validation | | | | | |
|---|---|---|---|---|---|
| **Objective** | **Signers 1 & 2** | **Signers 3 & 4** | **Signers 5 & 6** | **Signers 7 & 8** | **Signers 9 & 10** | **Average** |
| ERM | 54.43 (1.11) | 59.24 (0.23) | 48.89 (1.45) | 62.96 (1.25) | 65.62 (0.42) | 58.23 |
| IRM | 42.06 (1.17) | 43.16 (1.48) | 39.06 (1.39) | 46.22 (3.83) | 47.46 (1.95) | 43.59 |
| VREx | 46.29 (1.81) | 49.93 (0.45) | 42.84 (0.75) | 54.23 (0.59) | 56.90 (0.53) | 50.04 |
| GroupDRO | 50.52 (1.01) | 54.49 (1.87) | 45.12 (1.27) | 56.51 (1.66) | 63.22 (0.75) | 53.97 |
| IB-ERM | 56.51 (1.38) | 59.51 (0.91) | 51.82 (0.96) | 64.52 (0.61) | 66.54 (1.27) | 59.78 |
| SD | 56.58 (1.24) | 60.68 (1.08) | 49.35 (0.51) | 62.43 (0.83) | 64.06 (1.39) | 58.62 |

## C.8 HHAR

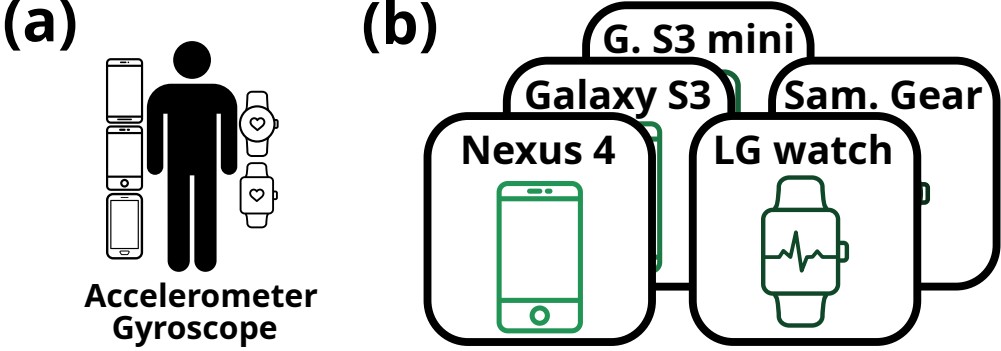

Figure 21: Summary of the HHAR dataset. (a) The task is to perform human activity classification from smart devices sensory data. (b) The dataset has five source domains, where each domain contains data gathered with a different smart device. The goal is to generalize to unseen smart devices.

### C.8.1 Setup

**Motivation** The intrinsic biases from inaccurate and poorly calibrated sensors of smart devices, along with the accumulated biases from everyday use makes human activity recognition a notoriously difficult task when task when done across devices (Stisen et al., 2015; Blunck et al., 2013). Contrary to static tasks where uninformative features can often be segmented out from the input features (e.g., background when classifying an animal from an image), invariant features in time series are often highly convoluted with other spurious features. We study the ability of models to ignore spurious information from complex signals with the HHAR (Stisen et al., 2015; Dua & Graff, 2017) dataset (Figure 21).

**Problem setting** We consider the human activity classification task from accelerometer and gyroscope measurements of smartphones and smartwatches. The dataset has five source domains, where each domain contains data gathered with a different device. The goal is to generalize to unseen smart devices.

**Data**  The dataset consists of 13674 recordings of 3-axis accelerometer and 3-axis gyroscope data from 5 different smart devices (3 smartphones and 2 smartwatches). The inputs **X** are five second recordings of a 6-dimensional signal sampled at 100Hz. The labels **Y** consist of 6 activities: Stand, Sit, Walk, Bike, Stairs up, and Stairs Down. Domains $d$ consist of five smart device models: Nexus 4, Galaxy S3, Galaxy S3 Mini, LG Watch, and Samsung Galaxy Gears.

**Preprocessing**  This section details the preprocessing steps taken for the HHAR dataset. The raw data was gathered with 10 different smart devices (2 from each model). Different models have different sampling frequencies, plus recordings have gaps in the data samples where devices temporarily stopped recording, making the time series irregularly sampled. We first remove the recordings of any device that either is missing considerable amounts of signals or has less than 100 seconds of recording. We then sort the data points in each sequence according to their recorded time, instead of time the data was saved on the device. Next, we split the full recordings into sequences of five seconds and resample at 100Hz. Finally, we normalize the data with a standard scaler applied to the accelerometer and gyroscope channels separately.

**Domain information**  Table 38 details the proportion of samples and labels across domains.

Table 38: Domain proportions of labels in the HHAR dataset

| Domain | Stand | Sit | Walk | Bike | Stairs up | Stairs down | Domain Total |
|---|---|---|---|---|---|---|---|
| Nexus 4 | 760 | 911 | 1024 | 644 | 695 | 543 | 4577 |
| Galaxy S3 | 664 | 889 | 944 | 560 | 635 | 474 | 4166 |
| Galaxy S3 Mini | 409 | 501 | 524 | 297 | 396 | 280 | 2407 |
| LG watch | 368 | 358 | 382 | 424 | 315 | 307 | 2154 |
| Gear watch | 21 | 23 | 78 | 42 | 120 | 86 | 370 |
| **Total** | 2222 | 2682 | 2952 | 1967 | 2161 | 1690 | 13674 |

**Architecture choice**  As this data is similar to EEG recordings, we use the same deep convolution network model as in the CAP and SEDFx datasets. The architecture is defined in work from Schirrmeister et al. (2017). We use the implementation of the BrainDecode (Schirrmeister et al., 2017) Toolbox. Temporal Convolutional Networks (TCN) are powerful tools for processing time series data (Bai et al., 2018). The architecture we use combines temporal and spatial convolution, which fits this data well. We found that it performed well on this task and obtained stable performance. The implementation is available at `https://github.com/TNTLFreiburg/braindecode`.

### C.8.2  Detailed results

**ID evaluation**  We show the results of ERM for the HHAR dataset in Table 39. We obtain these results by doing a hyperparameter search with the methodology detailed in Appendix F with no held-out test domain and choose the model with train-domain validation. In other words, the training is done with all domains; thus, all domains are ID. The columns correspond to the validation accuracy of the chosen model in each domain.

Table 39: ID results for the HHAR dataset

| Algorithm | Nexus 4 | Galazy S3 | Galaxy S3 Mini | LG watch | Sam. Gear | Average |
|---|---|---|---|---|---|---|
| ID ERM | 98.91 (0.24) | 98.44 (0.15) | 98.68 (0.15) | 90.08 (0.28) | 80.63 (1.33) | 93.35 |

**Benchmark results**  We show the detailed benchmark results of the adapted OOD generalization algorithms in Table 40. Each results is obtained by holding out one domain during training and reporting the performance of the chosen model from the hyperparameter sweep on that held out domain, more details in Appendix F.

Table 40: OOD generalization algorithms performance on the HHAR dataset

| Train-domain validation | | | | | | |
|---|---|---|---|---|---|---|
| **Objective** | Nexus 4 | Galazy S3 | Galaxy S3 Mini | LG watch | Sam. Gear | **Average** |
| ERM | 97.91 (0.03) | 98.17 (0.18) | 92.49 (0.26) | 71.33 (0.67) | 62.16 (1.69) | 84.41 |
| IRM | 95.68 (0.47) | 96.31 (0.53) | 91.10 (0.35) | 69.76 (1.44) | 61.71 (1.56) | 82.91 |
| VREx | 95.53 (0.55) | 96.51 (0.16) | 91.36 (0.43) | 69.72 (0.29) | 62.73 (1.15) | 83.17 |
| GroupDRO | 96.49 (0.18) | 96.79 (0.12) | 92.13 (0.09) | 71.64 (0.43) | 63.74 (1.34) | 84.16 |
| IB-ERM | 97.56 (0.06) | 97.93 (0.21) | 91.76 (0.57) | 71.38 (1.02) | 59.01 (1.86) | 83.53 |
| SD | 98.14 (0.01) | 98.32 (0.19) | 92.71 (0.09) | 75.12 (0.18) | 63.85 (0.28) | 85.63 |
| **Oracle train-domain validation** | | | | | | |
| **Objective** | Nexus 4 | Galazy S3 | Galaxy S3 Mini | LG watch | Sam. Gear | **Average** |
| ERM | 97.64 (0.06) | 98.05 (0.07) | 93.18 (0.20) | 73.11 (0.77) | 64.64 (1.20) | 85.32 |
| IRM | 96.81 (0.14) | 96.43 (0.09) | 91.26 (0.23) | 70.61 (0.51) | 61.82 (2.21) | 83.39 |
| VREx | 96.60 (0.24) | 96.68 (0.29) | 92.00 (0.65) | 71.67 (0.84) | 59.23 (1.17) | 83.24 |
| GroupDRO | 96.54 (0.23) | 96.94 (0.15) | 91.62 (0.34) | 71.33 (0.68) | 64.86 (0.69) | 84.26 |
| IB-ERM | 98.16 (0.09) | 98.22 (0.09) | 93.18 (0.16) | 73.40 (0.68) | 64.64 (0.09) | 85.52 |
| SD | 98.48 (0.01) | 98.67 (0.11) | 94.36 (0.24) | 75.12 (0.18) | 64.86 (0.28) | 86.30 |

### C.8.3 Credits and license

### C.9 PedCount

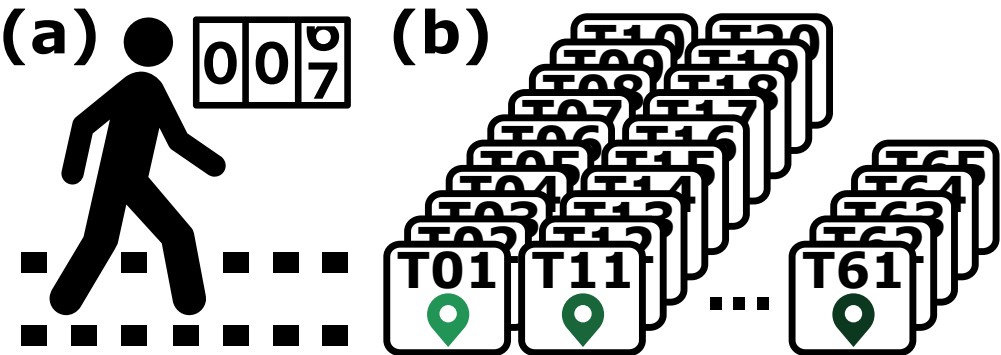

Figure 22: Summary of the PedCount dataset. (a) The task is to forecast the count of pedestrian crossing streets of Melbourne. (b) The dataset has 65 source domains, where each domain contains pedestrian counts of a different street crossing. The goal is to perform well on unseen street crossings.

### C.9.1 Setup

**Motivation**  Data gathered from the behavior of a population follows seasonal (daily, weekly, yearly) trends. An example of this is the movement of population within a city, either by walking, public transport or car. These trends form from the daily life of the population, e.g., the influx in the morning, outflux in the evening, and absence on the weekend. However, these trends can shift when the data is gathered from different sources in a city. We study the impact of those trend shifts with the Pedestrian (City of Melbourne, 2017; Godahewa et al., 2021) dataset (Figure 22).

**Problem setting**   The dataset has 65 source domains, where each domain contains pedestrian counts of a different street crossing. The goal is to perform well on unseen street crossings. Specifically, we investigate the OOD generalization to location T22 and T25.

**Data**   The dataset consists of 65 time series comprising pedestrian crossing counts in the city of Melbourne, Australia. The time series are gathered from various parts of the city. The time series are gathered up to 30/04/2020, and the start of the data gathering process range from 1/5/2009 to 13/3/2020. The inputs **X** are seven days of pedestrian count sampled hourly, to which we add 40 lag features and four time features. Lag features are past pedestrian count values that go past the seven day context given to the model. The time features are time indicators: hour of the day, day of the week, day of the month, and day of the year. The labels **Y** is the pedestrian count for the day following the seven days of context. Domains $d$ consist of 65 different counters (T1-T65).

**Preprocessing**   We do not perform any preprocessing for this dataset, this was already accomplished by prior work from Godahewa et al. (2021).

**Domain information**   Information on start date and end date of data gathering can be found in Table 41 along with some statistics such as time series average and maximum value.

**Architecture choice**   For this dataset, we use a forecasting Transformer architecture closely following the original formulation of Vaswani et al. (2017a). We found that it performed well on this task and obtained stable performance. Details are in Table 42.

Table 42: Model architecture used for the Pedestrian dataset

| # | Layer |
|---|-------|
| 1 | TransformerEncoder(d_model=48, nhead=2, num_encoder_layers=2, dim_feedforward=32, dropout=0.1, activation=gelu) |
| 2 | TransformerDecoder(d_model=48, nhead=2, num_encoder_layers=2, dim_feedforward=32, dropout=0.1, activation=gelu) |

### C.9.2   Detailed Results

**ID evaluation**   We show the in-distribution (ID) ERM results for the Pedestrian dataset in Table 43. We obtain these results by doing a hyperparameter search with the methodology detailed in Appendix F with no held-out test domain and choose the model with train-domain validation. In other words, the training is done with all domains; thus, all domains are ID. The columns correspond to the validation accuracy of the chosen model in each domain.

Table 43: ID results for the Pedestrian dataset

| **Algorithm** | T22 | T25 | T1-T65 \\{T22,T25} | **Average** |
|---|---|---|---|---|
| ID ERM | 96.40 (4.46) | 101.73 (1.02) | 61.48 (1.15) | 62.65 |

**Benchmark results**   We show the detailed benchmark results of the adapted OOD generalization algorithms in Table 44. Each results is obtained by holding out one domain during training and reporting the performance of the chosen model from the hyperparameter sweep on that held out domain, more details in Appendix F.

Table 41: Domain information in the Pedestrian dataset

| Domain | Start date | End date | Time series length | Time series average | Time series maximum |
|--------|-----------|----------|--------------------|---------------------|---------------------|
| T1 | 1/5/2009 | 13/12/2018 | 84331 | 1157 | 5573 |
| T2 | 1/5/2009 | 20/4/2020 | 96187 | 1074 | 7035 |
| T3 | 19/5/2009 | 30/10/2019 | 91594 | 1207 | 5890 |
| T4 | 1/5/2009 | 17/8/2019 | 90260 | 1480 | 8052 |
| T5 | 1/5/2009 | 12/12/2019 | 93068 | 1081 | 7391 |
| T6 | 1/5/2009 | 12/4/2020 | 95994 | 1193 | 6568 |
| T7 | 25/9/2009 | 15/8/2018 | 77924 | 366 | 11742 |
| T8 | 21/5/2009 | 14/3/2020 | 94801 | 151 | 3275 |
| T9 | 1/5/2009 | 30/4/2020 | 96424 | 518 | 5873 |
| T10 | 1/5/2009 | 12/4/2020 | 95985 | 176 | 3113 |
| T11 | 1/5/2009 | 27/2/2020 | 94908 | 99 | 9805 |
| T12 | 1/5/2009 | 6/12/2019 | 92917 | 202 | 11284 |
| T13 | 1/5/2009 | 26/4/2017 | 70028 | 743 | 7510 |
| T14 | 1/5/2009 | 14/9/2019 | 90930 | 398 | 7304 |
| T15 | 1/5/2009 | 13/6/2019 | 88700 | 800 | 5559 |
| T16 | 1/5/2009 | 3/7/2014 | 45359 | 713 | 4640 |
| T17 | 1/5/2009 | 10/8/2019 | 90091 | 460 | 3938 |
| T18 | 1/5/2009 | 30/4/2020 | 96423 | 344 | 3759 |
| T19 | 1/9/2013 | 1/3/2020 | 56969 | 566 | 2544 |
| T20 | 6/9/2013 | 20/3/2020 | 57307 | 372 | 2231 |
| T21 | 1/9/2013 | 12/4/2020 | 57979 | 606 | 5438 |
| T22 | 1/9/2013 | 1/12/2018 | 46030 | 1531 | 5654 |
| T23 | 1/9/2013 | 30/1/2020 | 56227 | 334 | 3845 |
| T24 | 1/9/2013 | 10/3/2020 | 57187 | 1195 | 5880 |
| T25 | 1/9/2013 | 3/12/2019 | 54826 | 561 | 7664 |
| T26 | 28/9/2013 | 27/4/2020 | 57691 | 548 | 2957 |
| T27 | 1/9/2013 | 20/3/2020 | 57426 | 127 | 888 |
| T28 | 20/9/2013 | 19/3/2020 | 56946 | 984 | 7954 |
| T29 | 11/10/2013 | 16/12/2019 | 54186 | 444 | 7494 |
| T30 | 16/10/2013 | 13/12/2019 | 53996 | 506 | 2801 |
| T31 | 9/10/2013 | 15/4/2020 | 57139 | 282 | 3040 |
| T32 | 20/12/2013 | 25/12/2016 | 26443 | 1057 | 9791 |
| T33 | 23/4/2014 | 15/9/2019 | 47315 | 156 | 2218 |
| T34 | 8/6/2014 | 30/4/2020 | 51683 | 144 | 3708 |
| T35 | 12/4/2016 | 25/4/2020 | 35398 | 1545 | 9912 |
| T36 | 21/1/2015 | 19/1/2020 | 43795 | 291 | 1389 |
| T37 | 1/2/2015 | 30/4/2020 | 45976 | 159 | 1656 |
| T38 | 1/1/2015 | 5/1/2017 | 17662 | 2448 | 6965 |
| T39 | 23/8/2014 | 10/11/2019 | 45724 | 226 | 1823 |
| T40 | 20/1/2015 | 9/4/2020 | 45765 | 287 | 2103 |
| T41 | 1/7/2017 | 20/4/2019 | 15815 | 1786 | 7138 |
| T42 | 15/4/2015 | 24/3/2020 | 43336 | 247 | 2292 |
| T43 | 15/4/2015 | 30/4/2020 | 44219 | 201 | 1539 |
| T44 | 15/4/2015 | 29/4/2020 | 44207 | 97 | 906 |
| T45 | 1/7/2017 | 17/12/2019 | 21599 | 883 | 4736 |
| T46 | 8/8/2017 | 12/4/2020 | 23472 | 100 | 678 |
| T47 | 25/8/2017 | 26/4/2020 | 23411 | 950 | 4532 |
| T48 | 3/10/2017 | 1/4/2020 | 21876 | 249 | 2375 |
| T49 | 30/11/2017 | 14/4/2020 | 20784 | 181 | 2167 |
| T50 | 1/7/2017 | 4/4/2020 | 24215 | 255 | 2271 |
| T51 | 1/12/2017 | 1/5/2020 | 21168 | 124 | 564 |
| T52 | 1/8/2017 | 25/4/2020 | 23974 | 395 | 1918 |
| T53 | 1/10/2015 | 13/4/2020 | 39765 | 695 | 3738 |
| T54 | 1/7/2018 | 29/4/2020 | 16032 | 148 | 1316 |
| T55 | 1/8/2018 | 23/3/2019 | 5616 | 817 | 2638 |
| T56 | 1/8/2018 | 18/4/2020 | 15027 | 310 | 1219 |
| T57 | 1/9/2018 | 11/3/2020 | 13368 | 798 | 15979 |
| T58 | 1/10/2018 | 1/5/2020 | 13872 | 745 | 3352 |
| T59 | 13/2/2019 | 1/5/2020 | 10632 | 252 | 3849 |
| T60 | 18/4/2019 | 23/7/2019 | 2291 | 1600 | 5424 |
| T61 | 1/7/2019 | 1/5/2020 | 7320 | 447 | 2984 |
| T62 | 1/10/2019 | 1/5/2020 | 5112 | 120 | 606 |
| T63 | 8/1/2020 | 22/3/2020 | 1777 | 294 | 2106 |
| T64 | 17/1/2020 | 14/4/2020 | 2112 | 168 | 1157 |
| T65 | 13/3/2020 | 1/5/2020 | 1176 | 136 | 1486 |
| **Total** | 2222 | 2682 | 2952 | 1967 | 2161 |

### C.9.3   Credits and license

This dataset was adapted from the work of City of Melbourne (2017) as made available on the online Monash time series archive (Godahewa et al., 2021). This dataset is licensed under the Creative Commons Attribution 4.0 International License.

Table 44: OOD generalization algorithms performance on the Pedestrian dataset

| Train-domain validation | | | |
| --- | --- | --- | --- |
| Objective | T22 | T25 | Average |
| ERM | 196.07 (10.21) | 212.11 (12.65) | 204.09 |
| VREx | 197.98 (7.31) | 205.19 (4.74) | 201.58 |
| GroupDRO | 243.53 (16.90) | 242.94 (9.17) | 243.23 |
| IB-ERM | 224.55 (15.43) | 201.60 (6.41) | 213.07 |
| Oracle train-domain validation | | | |
| Objective | T22 | T25 | Average |
| ERM | 226.78 (11.88) | 219.71 (2.25) | 223.24 |
| VREx | 203.52 (2.82) | 222.63 (4.39) | 213.07 |
| GroupDRO | 261.10 (12.11) | 223.04 (7.84) | 242.06 |
| IB-ERM | 201.43 (10.94) | 209.89 (11.65) | 205.66 |

## C.10 AusElec

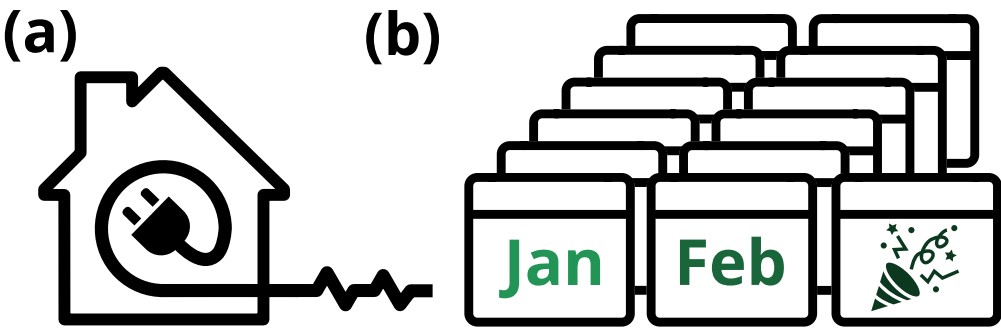

Figure 23: Summary of the AusElec dataset. (a) The task is to forecast electricity consumption. (b) The dataset has 13 time domains, where each domain contains data from different months and holidays. The goal is to perform well on all seasonalities.

### C.10.1 Setup

**Motivation** Seasonality is the property of time series where recurring characteristics appear every cycle of a fixed period, e.g., weekly. A common practice in the forecasting field is to provide models with additional information, e.g., day of week in order to allow models to leverage seasonality for better predictions. However, holidays is a seasonality of time series that is very sparse which models often fail to capture. We study the performance of models on sparse seasonality with the AusElec (Hyndman & Athanasopoulos, 2018; Godahewa et al., 2021) dataset (Figure 23)

**Problem setting** We consider the electricity consumption forecasting task. The dataset has 13 time domains, where each domain contains data from different months and holidays. The goal is to perform well on all seasonalities.

**Data** The dataset consists of five time series comprising 13 years of electricity demand across five states in Australia: Victoria, New South Wales, Queensland, Tasmania and South Australia. The inputs **X** are seven days of electricity demand sampled half hourly to which we add 42 lag features and 5 time features. Lag features are past electricity demand values the goes past the seven day context given to the model. The time features are time indicators: minute of hour, hour of day, day of week, day of month, and day of year.

The labels **Y** is the electricity demand for the day following the seven days of context. Domains $d$ consist of time intervals throughout the year: January, February, March, April, May, June, July, August, September, October, November, December, and holidays.

**Preprocessing**  We do not perform any preprocessing for this dataset, this was already accomplished by prior work from Godahewa et al. (2021).

**Domain information**  We define the time interval of the Holidays domain as union of the following Australian holidays: New Year's Day, Australia Day, Good Friday, Easter Monday, Anzac Day, Christmas Day, Boxing Day.

**Architecture choice**  For this dataset, we use a forecasting Transformer architecture closely following the original formulation of Vaswani et al. (2017a). We found that it performed well on this task and obtained stable performance. Details are in Table 45.

Table 45: Model architecture used for the AusElec dataset

| # | Layer |
|---|---|
| 1 | TransformerEncoder(d_model=48, nhead=2, num_encoder_layers=2, dim_feedforward=32, dropout=0.1, activation=gelu) |
| 2 | TransformerDecoder(d_model=48, nhead=2, num_encoder_layers=2, dim_feedforward=32, dropout=0.1, activation=gelu) |

### C.10.2  Detailed Results

**Unbalanced results**  It has been reported in prior work (Koh et al., 2021) that OOD generalization algorithms such as IRM outperforms ERM on subpopulation shift datasets. However, it is unclear whether the improvements originates from the nature OOD generalization algorithms to upsample minority domains when computing the empirical risk or because the algorithm is performing well. In this work, we create an Unbalanced dataset of the subpopulation shift dataset which is agnostic of the domain definition during training. This allows us to compare the gain in performance obtained by upsampling the minority domain when minimizing the empirical risk. We show those results in Table 46.

Table 46: Results for the AusElecUnbalanced dataset

| Average validation | | | Worst-domain validation | | |
|---|---|---|---|---|---|
| **Objective** | Average | **Worse** | **Objective** | Average | **Worse** |
| ERM | 227.73 (2.64) | 409.80 (4.21) | ERM | 235.40 (4.38) | 395.99 (5.49) |

**Benchmark results**  We show the detailed benchmark results of the adapted OOD generalization algorithms in Table 47. Each line is obtained by training on all domains of the dataset and reporting the average and worst domain performance of the chosen model, more details in Appendix F.

### C.10.3  Credits and license

This dataset was adapted from the work of Hyndman & Athanasopoulos (2018) as made available on the online Monash time series archive (Godahewa et al., 2021). This dataset is licensed under the Creative Commons Attribution 4.0 International License.

Table 47: OOD generalization algorithms performance on the AusElectricity dataset

| Average validation | | | Worst-domain validation | | |
|---|---|---|---|---|---|
| **Objective** | Average | **Worse** | **Objective** | Average | **Worse** |
| ERM | 232.01 (2.60) | 397.27 (8.48) | ERM | 247.08 (7.59) | 403.56 (6.57) |
| VREx | 237.96 (2.53) | 415.01 (9.92) | VREx | 247.09 (2.19) | 408.87 (3.97) |
| GroupDRO | 237.09 (3.63) | 408.83 (2.37) | GroupDRO | 252.95 (7.58) | 424.44 (13.34) |
| IB-ERM | 232.03 (2.68) | 393.56 (2.41) | IB-ERM | 235.87 (3.11) | 391.13 (5.44) |

### C.11 IEMOCAP

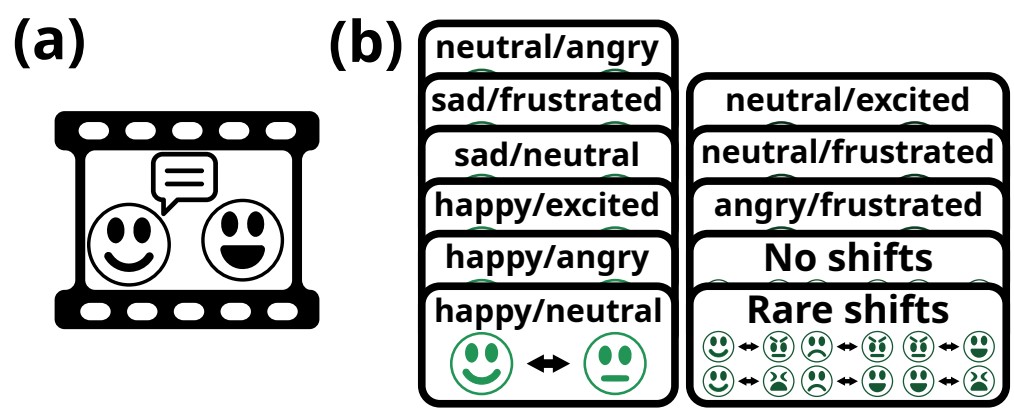

Figure 24: Summary of the IEMOCAP dataset. (a) The task is to perform emotion recognition from multi modal data (video, sound, text). (b) The dataset has 11 time domains, where each domain contains data from a different emotion shifts during conversations. The goal is to perform well on all conversational emotion shifts.

#### C.11.1 Setup

**Motivation**   Speakers tend to maintain an emotional state over a conversation. However, external stimuli can invoke a shift in the emotional state of speakers (Poria et al., 2019). Such emotion shift are often sparsely represented in the data, making it hard for models to classify them adequately. Recent work on emotion recognition models (Poria et al., 2019; 2018; Majumder et al., 2019) show the failure of existing models to adapt to those emotion shift. We study the performance of models on emotional shift with the IEMOCAP (Bulut et al., 2008) dataset (Figure 12).

**Problem setting**   We consider the emotion recognition task. The dataset has 11 time domains, where each domain contains data from a different emotion shift during conversations. The goal is to perform well on all conversational emotion shifts.

**Data**   The dataset consists of 151 videos about dyadic interactions, where professional actors are required to perform scripted scenes that elicit specific emotions. Each video contains a single dyadic dialogue, segmented into utterances. It contains 7433 utterances in total. The inputs $\mathbf{X}$ are utterances of video, speech, and text transcriptions. The labels $\mathbf{Y}$ consist of 6 emotions: Happy, Sad, Neutral, Angry, Excited, and Frustrated. Domains $d$ consist of 11 emotion shift during conversations: No-Shift, Rare-Shift, and 9 common emotion shifts including Happy-Neutral, Happy-Angry, Happy-Excited, Sad-Neutral, Sad-Frustrated, Neutral-Angry, Neutral-Excited, Neutral-Frustrated, and Angry-Frustrated .

**Preprocessing**   This section details the preprocessing steps taken to bring the IEMOCAP dataset from its raw form to its final form used in WOODS. For each utterance, we extract multimodal features (audio, visual

and text) following the same approach as Hazarika et al. (2018) and Majumder et al. (2019). To get our text embedding, we use a simple CNN with one convolutional layer followed by max-pooling (Poria et al., 2016). To extract high dimensional audio vectors, we use openSMILE (Eyben et al., 2010). These vectors comprise features like loudness, Mel-spectra, MFCC, pitch, etc. We use a 3D-CNN to capture video embeddings (Tran et al., 2015). This embedding contains information for detecting emotional expressions like a smile or frown. We use concatenation of the unimodal features as a fusion method.

**Domain information**   We consider utterances that have the same label as the previous utterance spoken by the same speaker as a no-shift domain. We consider emotion-shifts that appear in less than 20 utterances as rare-shift domain, namely, Happy-Angry, Excited-Angry, Frustrated-Happy, Sad-Excited, frustrated-Excited, and Sad-Angry. We consider the remaining 9 emotion shifts as common ones and create a separate domain for each of them. For brevity, we call these domains common-shift in general. The ratios for the rare emotion-shift domain are 1/6, 1/6, and 2/3 for training, validation, and test respectively. For the remaining domains, dialogues are randomly chosen to achieve the ratios of 0.7, 0.1, and 0.2 for the size of training, validation, and test respectively.

Table 48 details the proportion of utterances and dialogues in the training, validation, and test sets across domains.

Table 48: Domain proportions of utterances and dialogues in the training, validation and test sets of IEMOCAP dataset

|                                               | Training | Validation | Test |
|-----------------------------------------------|----------|------------|------|
| # of utterances in rare-shift domain          | 22       | 19         | 61   |
| # of utterances in no-shift domain            | 3785     | 369        | 957  |
| total # of utterance in common-shift domains  | 1297     | 196        | 527  |
| **total # of utterances**                     | 5298     | 589        | 1546 |
| **total # of dialogues**                      | 108      | 12         | 31   |

**Architecture choice**   For this dataset, we use a DialogueRNN model as defined in work from Majumder et al. (2019). We chose this model because it is well recognized by the ERM community. It also has an effective mechanisms to model context by tracking individual speaker states throughout the conversation for emotion classification. The implementation is available at `https://github.com/declare-lab/conv-emotion/tree/master/DialogueRNN`.

### C.11.2   Detailed results

**Unbalanced results**   It has been reported in prior work (Koh et al., 2021) that OOD generalization algorithms such as IRM outperforms ERM on subpopulation shift datasets. However, it is unclear whether the improvements originates from the nature OOD generalization algorithms to upsample minority domains when computing the empirical risk or because the algorithm is performing well. In this work, we create an Unbalanced dataset of the subpopulation shift dataset which is agnostic of the domain definition during training. This allows us to compare the gain in performance obtained by upsampling the minority domain when minimizing the empirical risk. We show those results in Table 49.

Table 49: Results for the IEMOCAPUnbalanced dataset

| Average validation | | | Worst-domain validation | | |
|---|---|---|---|---|---|
| **Objective** | Average | **Worst** | **Objective** | Average | **Worst** |
| ERM | 70.53 (0.05) | 58.24 (1.41) | ERM | 70.01 (0.77) | 56.76 (1.24) |

**Benchmark results**    We show the detailed benchmark results of the adapted OOD generalization algorithms in Table 50. Each line is obtained by training on all domains of the dataset and reporting the average and worst domain performance of the chosen model, more details in Appendix F.

Table 50: OOD generalization algorithms performance on the IEMOCAP dataset

| Average validation | | | Worst-domain validation | | |
|---|---|---|---|---|---|
| **Objective** | Average | **Worst** | **Objective** | Average | **Worst** |
| ERM | 69.12 (0.36) | 57.75 (1.85) | ERM | 69.85 (0.03) | 56.33 (2.76) |
| IRM | 68.73 (0.24) | 55.93 (1.20) | IRM | 70.21 (0.31) | 58.95 (1.13) |
| VREx | 70.12 (0.51) | 59.45 (1.43) | VREx | 69.64 (0.44) | 57.66 (3.13) |
| GroupDRO | 69.21 (0.75) | 56.11 (1.19) | GroupDRO | 70.08 (0.86) | 58.79 (1.00) |
| IB-ERM | 68.79 (0.08) | 59.93 (0.55) | IB-ERM | 70.04 (0.42) | 58.81 (1.50) |
| SD | 68.62 (0.22) | 58.04 (0.39) | SD | 68.75 (0.28) | 56.14 (1.24) |

### C.11.3   Credits and license

This dataset was adapted from the work of Bulut et al. (2008) as made available by the Speech Analysis and Interpretation Laboratory (SAIL) at the University of Southern California (USC). This dataset is licensed under the license availabel at the license at `https://sail.usc.edu/iemocap/iemocap_release.htm`.

## D    Further details on adapation of OOD generalization algorithms

### D.1   General adaptation of OOD generalization algorithms to time series

The problem formulation in Section 2.2 applies only sequence of same length $S_t$ and prediction times $S_p$ across samples. However, for several dataset and tasks, this does not hold up. Take as example the IEMOCAP dataset, conversations can vary in length and prediction times across samples. In this section, we provide a general formulation that accounts these changes.

Data samples consist of the input time series observation $\mathbf{X}^i = [X_t^i]_{t \in S_t^i}$, where $S_t^i$ is the set of time steps for sample $i$, and the set of labels $\mathbf{Y}^i = [Y_t^i]_{t \in S_p^i}$, where $S_p^i \subseteq S_t^i$ is the set of labeled time steps for sample $i$.

**Empirical risk**    For the empirical risk of domain $d$, we average the risk across the set of labeled time steps of sample $i$ belonging to domain $d$: $S_p^{d,i}$.

$$R^d(f) = \frac{1}{n^d} \sum_{(\mathbf{X}^i, \mathbf{Y}^i) \in D} \frac{1}{|S_p^{d,i}|} \sum_{t \in S_p^{d,i}} \mathcal{L}\big(f(X_{1:t}^i), Y_t^i\big), \tag{3}$$

where $n^d$ is the number of samples from domain $d$ in the dataset $D$.

**Penalty value function**    IB-ERM and SD penalize representation and logits during prediction, we follow Equation (1) and define the penalty below.

$$P(f) = \frac{1}{n^d} \sum_{(\mathbf{X}^i, \mathbf{Y}^i) \in D} \frac{1}{|S_p^{d,i}|} \sum_{t \in S_p^{d,i}} \tilde{P}(f, X_{1:t}^i, Y_t^i). \tag{4}$$

### D.2   OOD generalization algorithm definition

- **IRM** performs a constrained empirical risk minimization such that the optimal classifier of representations is the same across the domains. It does so by penalizing a function of empirical risk across

domains. We adapt IRM by using the empirical risk from Equation (3):

$$P(f) = \frac{1}{d} \sum_{D^d \in D} \|\nabla_{w|w=1.0} R^d(w \cdot f)\|^2, \tag{IRM}$$

where $|d|$ is the number of domains.

- **VREx** penalizes the variance of risk across domains We adapt VREx by using the empirical risk from Equation (3):

$$P(f) = \mathsf{Var}_{D^d \in D}\big(R^d(f)\big), \tag{VREx}$$

where $\mathsf{Var}$ is the variance taken across domains.

- **GroupDRO** performs importance weighting of the domains when calculating the empirical risk. We adapt the domain weighting parameter $\mathbf{q}_d$ using the empirical risk from Equation (3):

$$\mathbf{q}_d = \frac{\mathbf{q}_d' e^{R^d(f)}}{\sum_{D^d \in D} \mathbf{q}_d'(f)}, \tag{GroupDRO}$$

where $\mathbf{q}_d'$ is the domain weights from the previous iteration.

- **IB-ERM** penalizes the variance of representation within domains. Consider a representation map $\Phi$ (that transforms inputs $\mathbf{X}$ as $\Phi(\mathbf{X})$) and a linear classifier $w$ such that our predictor $f$ is defined as $w \cdot \Phi$. We define the IB-ERM penalty as:

$$P(f) = \frac{1}{|d|} \sum_{D^d \in D} \mathsf{Var}_{(\mathbf{X},\mathbf{Y}) \in D^d}\big(\Phi(\mathbf{X})\big), \tag{IB-ERM}$$

where $\mathsf{Var}$ is the variance is taken across samples of a domain.

- **SD** penalizes the squared l2 norm of the logits of the predictor $f$:

$$P(f) = \frac{1}{n^d} \sum_{(\mathbf{X}^i, \mathbf{Y}^i) \in D} \frac{1}{|S_p^{d,i}|} \sum_{t \in S_p^{d,i}} \|f(X_{1:t}^i)\|^2, \tag{SD}$$

where $n^d$ is the number of samples from domain $d$ in the dataset $D$.

# E   Measuring the impact of distribution shifts

We use the *generalization gap* to empirically measure the impact of the distribution shifts on the performance of models. It measures the drop in performance between data drawn In-Distribution (ID) and Out-of-Distribution (OOD), where the former is independent and identically distributed (i.i.d.) to the training distribution and the later is not. [2] However, the generalization gap can be a misleading measure as it does not intrinsically indicate attainable performance gains. We show an example of unattainable performance gains later in this section. In this work, we do our best to measure an achievable performance gap for our dataset, i.e., an upper bound to the achievable performance on unseen domains. In this section we give details on of the generalization gaps described in Table 1 are obtained for domain generalization and subpopulation shift datasets.

## E.1   Generalization gap for domain generalization

Given a set of training domains $D^{\mathsf{train}}$ and a test domain $D^{\mathsf{test}}$, we measure the OOD performance by training a model on the training domains $D^{\mathsf{train}}$ with ERM and measure the performance of this model on the test domain $D^{\mathsf{test}}$. The ID performance can be measured in multiple ways. Koh et al. (2021) provides multiple definitions for it:

---

[2]Some restriction with respect to the training distribution is implied, see Section 2.1.

- **Train-to-train** Performance of a model on $D^{\text{train}}$ when trained on $D^{\text{train}}$

- **Mixed-to-test** Performance of a model on $D^{\text{test}}$ when trained on a mixture of $D^{\text{train}}$ and $D^{\text{test}}$.

- **Test-to-test** Performance of a model on $D^{\text{test}}$ when trained on $D^{\text{test}}$

We use the mixed-to-test measure for ID performance, because test-to-test and train-to-train can lead to erroneous measures leading to an inflated generalization gap that is unattainable in reality. To illustrate this problem, consider the generalization gap obtained with the train-to-train ID performance on the Spurious-Fourier dataset.

**Example E.1** (Unattainable performance gap)**.** Performing ERM on the training domains $D^d|_{d \in \{80\%, 90\%\}}$ will lead to a model relying on the spurious features to make predictions, as they are a stronger predictor of the label (85%) than the invariant features (75%). Thus, the model will achieve 85% accuracy on data sampled ID to domains $d \in \{80\%, 90\%\}$, but only achieve 10% accuracy on the test domain $D^{10\%}$. Comparing the ID and OOD performance would lead to a generalization gap of 75%. However, this gap is misleading as a model could never achieve 85% accuracy on the test domain because the strongest invariant predictor can only achieve 75%. A similar case can be made for the test-to-test measure of ID performance, where the generalization gap lead to an unattainable performance on the test domain.

Instead, consider the generalization gap obtained with the mixed-to-test ID performance for the same dataset.

**Example E.2** (Attainable performance gap)**.** Performing ERM on the training domains $D^d|_{d \in \{10\%, 80\%, 90\%\}}$ will lead to a predictor that relies on the invariant features, as they are a stronger predictor of the label (75%) than the spurious features (60%). Therefore, the ID performance will be 75%, and the OOD performance will be 10%, leading to a generalization gap of 65%. This gap is a much more significant measure of the upper bound of the performance than the original definition.

To summarize, we compute the generalization gap for domain generalization datasets as follows. Given a set of training domains $D^{\text{train}}$ and a test domain $D^{\text{test}}$, we first measure the *OOD performance* by training a model on the training domains $D^{\text{train}}$ with ERM and measure the performance of this model on the test domain $D^{\text{test}}$. Second, we measure the *ID performance* by training a model on all domains $D = \cup_{d=\{\text{train,test}\}} D^d$ and evaluate the model on the test domain $D^{\text{test}}$. The generalization gap for that test domain is then defined as the difference between the ID and OOD performance. That process can then be repeated for all domains in the dataset and we average the performance.

### E.2 Generalization gap for subpopulation shifts

In subpopulation shift datasets, we measure the OOD performance as the worst domain performance, and the ID performance as the train-to-train performance, i.e., the average domain performance. We recognize that this measure is not a perfect because of similar arguments made in Section E.1, e.g., one domain might be much more difficult that the others and thus might be impossible to achieve the level of average performance on it. However, we argue that it is reasonable to consider the gap between the average domain performance and the worst domain performance as attainable. As a sanity check, one could verify that the average performance is achievable for a domain $d$ by doing a test-to-test style measure. We leave this to future work.

To summarize, we compute the generalization gap for subpopulation shifts datasets as follows. Given a set of training domains $D^{\text{train}}$. we measure the ID performance by training a model on all domains, and measure the average performance across domains. We define the OOD performance as the performance of the model on the worst domain in $D^{\text{train}}$. The generalization gap for that dataset is then defined as the different between the ID and OOD performance.

## F   Evaluation framework workflow

In this section of the appendix, we detail the methodology employed to evaluate the performance of OOD generalization algorithms on our datasets. We follow the workflow used by Gulrajani & Lopez-Paz (2020) in their DomainBed testbed and adapt the framework to time series tasks.

### F.1 Reported performance

We detail in this section the performance measure used for the different datasets in the WOODS benchmark.

**Synthetic challenge datasets**  Spurious-Fourier, TCMNIST-Source and TCMNIST-Time were formulated to address specific OOD generalization challenges in time series; thus we only investigate the training and testing domain configuration of interest, i.e., $D^{\mathsf{train}} = D^d|_{d \in \{80\%, 90\%\}}$ and $D^{\mathsf{test}} = D^{10\%}$. With this domain configuration, we perform a hyperparameter sweep, the model selection (see Section G) and report the performance of the chosen model on the 10% domain.

**Real-world domain generalization datasets**  We report the performance of an OOD generalization algorithm with a domain cross-validation measure as follows. For every domain in a dataset, we perform a hyperparameter sweep with that domain held out from training. After this hyperparameter search, we perform the model selection associated with the dataset (see Section G) and report the performance of the chosen model on the held out test set. We then report the average performance across domains.

**Real-world subpopulation shift datasets**  We report the performance of an OOD generalization algorithm with the worst domain performance. We perform a hyperparameter sweep with all domains in the training dataset $D^{\mathsf{train}}$. After this search, we perform model selection (see Section G) and report the worst domain performance.

### F.2 Systematic framework

**Hyperparameter search**  All hyperparameter searches in this work use random searches (Bergstra & Bengio, 2012) over the hyperparameter distribution spaces defined in Table 51 and Table 52. We train 20 models using randomly sampled hyperparameter configurations. We then select the best performing model of those 20 configurations using different validation sets definitions, see Appendix G.

**Statistically relevant**  We repeat each hyperparameter search three times to obtain statistically relevant results. This reduces the probability that some algorithm samples a lucky configuration of hyperparameters. All the results reported in this work are averaged over those three trials with a different seed. We also provide the estimated standard deviation of those averaged results.

**Reducing bias**  The search range is an important topic when discussing the fairness of this evaluation strategy. Having reasonable hyperparameter distributions for sampling in the random search is essential to remaining fair between the algorithms and reducing the induced bias in the results. Defining a narrow hyperparameter distribution for which one knows the algorithm performs very well on a dataset or test domain leads to a bias of the evaluation due to queries of the test domain through human intervention. This bias could lead to algorithms getting better results by increasing the chance of the random search finding a good value. When defining the hyperparameter range, one should define a range wide enough as to cover at least the relevant search space for this hyperparameter. In this work we use ranges that accurately reflects the range of useful hyperparameters values, see Table 52.

## G  Model Selection

Section F detailed the hyperparameter search and uncertainty estimation used in this framework. In this section, we detail the model selection strategy used in hyperparameter sweeps to determine the model to evaluate on the test domain.

### G.1 Model selection for domain generalization

A fundamental restriction in domain generalization is that the training procedure does not have access to the test domains during training. As a result, the challenge of OOD generalization is not only to create models that generalize to the test domains but also to select the right models without having access to the test

Table 51: Distributions of training hyperparameters for random search

| Dataset | Hyperparameter | Random distribution |
|---|---|---|
| Spurious-Fourier | learning rate
batch size
class balance | $10^{\text{Uniform}(-4.5,-2.5)}$
$2^{\text{Uniform}(3,9)}$
True |
| TCMNIST-Source | learning rate
batch size
class balance | $10^{\text{Uniform}(-4.5,-2.5)}$
$2^{\text{Uniform}(3,9)}$
True |
| TCMNIST-Time | learning rate
batch size
class balance | $10^{\text{Uniform}(-4.5,-2.5)}$
$2^{\text{Uniform}(3,9)}$
True |
| CAP | learning rate
batch size
class balance | $10^{\text{Uniform}(-5,-3)}$
$2^{\text{Uniform}(3,4)}$
True |
| SEDFx | learning rate
batch size
class balance | $10^{\text{Uniform}(-5,-3)}$
$2^{\text{Uniform}(3,4)}$
True |
| PCL | learning rate
batch size
class balance | $10^{\text{Uniform}(-5,-3)}$
$2^{\text{Uniform}(3,5)}$
True |
| LSA64 | learning rate
batch size
class balance | $10^{\text{Uniform}(-5,-3)}$
$2^{\text{Uniform}(3,4)}$
True |
| HHAR | learning rate
batch size
class balance | $10^{\text{Uniform}(-4,-2)}$
$2^{\text{Uniform}(3,4)}$
True |
| PedCount | learning rate
batch size
class balance | $10^{\text{Uniform}(-5,-3)}$
$2^{\text{Uniform}(3,5)}$
True |
| AusElec | learning rate
batch size
class balance | $10^{\text{Uniform}(-5,-3)}$
$2^{\text{Uniform}(3,5)}$
True |
| IEMOCAP | learning rate
batch size
class balance | $10^{\text{Uniform}(-5,-3)}$
$2^{\text{Uniform}(1,4)}$
True |

domains. Many model selection strategies were proposed (Gulrajani & Lopez-Paz, 2020; Ye et al., 2021b; Koh et al., 2021), the simplest of which is *Train-domain validation.*

**Train-domain validation** We split the training domains into training and validation sets. The training split of the training domain is used to train the model. We choose the model that gets the best average validation performance across training domains. We report the performance of the chosen model on the testing domains.

However, tackling both problems of creating and finding invariant models at the same time might be a very difficult research endeavor. Instead, we can first start by narrowing the scope and only focus on creating invariant models. For this purpose, we relax the fundamental restriction and allow the queries of the test

Table 52: Distributions of algorithm hyperparameters for random search

| Dataset | Hyperparameter | Random distribution |
|---|---|---|
| Invariant Risk Minimization | penalty weight
annealing iterations | $10^{\mathsf{Uniform}(-1,5)}$
$\mathsf{Uniform}(0, 2000)$ |
| Variational REx | penalty weight
annealing iterations | $10^{\mathsf{Uniform}(-1,5)}$
$\mathsf{Uniform}(0, 2000)$ |
| GroupDRO | $\eta$ | $10^{\mathsf{Uniform}(-3,-1)}$ |
| IB-ERM | penalty weight | $10^{\mathsf{Uniform}(-3,0)}$ |
| Spectral Decoupling | penalty weight | $10^{\mathsf{Uniform}(-5,-1)}$ |
| CAD | $\lambda$
temperature | $\mathsf{Choice}([10^{-4}, 10^{-3}, 10^{-2}, 10^{-1}, 1, 10^1, 10^2])$
$\mathsf{Choice}([0.05, 0.1])$ |
| CondCAD | $\lambda$
temperature | $\mathsf{Choice}([10^{-4}, 10^{-3}, 10^{-2}, 10^{-1}, 1, 10^1, 10^2])$
$\mathsf{Choice}([0.05, 0.1])$ |
| Transfer | $\lambda$
$\delta$
adv lr
adv steps | $10^{\mathsf{Uniform}(-2,1)}$
$\mathsf{Uniform}(0.1, 3.0)$
$10^{\mathsf{Uniform}(-4.5,-2.5)}$
$\mathsf{Choice}([1, 2, 5])$ |
| CCDG | $\alpha$
temperature | $\mathsf{Uniform}(0, 1)$
$\mathsf{Uniform}(0, 1)$ |
| Diversify | $\lambda_1$
$\lambda_2$ | $\mathsf{Uniform}(0, 1)$
$\mathsf{Uniform}(0, 1)$ |

domain to obtain some signal on the absolute performance of an algorithm. Although querying the test domain can never be considered a valid model selection strategy in practical scenarios, the results can be very insightful when evaluating the behavior of an algorithm. Gulrajani & Lopez-Paz (2020) formulated *Test-domain validation* that queries the test domains to perform model selection.

**Test-domain validation**  We split the test domains into testing and validation sets. Models are trained for a fixed number of training steps on the training domains. We choose the model with the best performance on the validation set of the test domains. However, we only consider the final checkpoint of the model after a fixed number of steps, effectively disallowing early stopping. We report the performance of the chosen model on the testing set of test domains.

Test-domain validation has proven to be a very useful measure of performance for algorithms on synthetic datasets driven primarily by correlation shift (Ye et al., 2021b), e.g., CMNIST. In such datasets, simple spurious features highly correlated with the label create shortcuts in the data that model leverage to minimize the empirical risk quickly (e.g., cow or camel classification problem). As a result, these shortcuts lead to very high training domain performance and very low test domain performance early in training. Consequently, any model selection criteria that rely on performance on data drawn i.i.d. to the training distribution is a poor way to investigate the performance of an algorithm because there is a bias of model selection towards early training correlation. Thus, by disallowing early stopping, we obtain an insightful measure to investigate the absolute performance of an algorithm.

On the other hand, Test-domain validation is ill-equipped to provide meaningful measures of performance with other kinds of datasets. For example, Test-domain validation is not an insightful measure of performance when dealing with real-world datasets. The reason is that we often do not know beforehand the number of training steps required for a given set of hyperparameters such that a model will finish the learning of the task. Therefore, we train models past the point of overfitting and pick the model with the highest validation

performance. This renders the last checkpoint in training suboptimal for generalization performance, both ID and OOD, and leads to an uninformative measure of the generalization performance.

We introduce a more pragmatic model selection method that queries the test domain for real-world datasets to resolve this problem: *Oracle train-domain validation.*

**Oracle train-domain validation**   We split the training domains into training and ID validation splits. We also split the test domains into testing and OOD validation splits. For every model training run, we choose the early stopped model that performs best on the ID validation split. Among all early stopped model of the sweep, we then choose the model that performs the best in the OOD validation split. Notice that this model selection method has the same number of queries of the test domain as the test-domain validation, i.e., one query per training run.

In light of the discussion of this section, we use two different sets of model selection methods for the two different types of datasets in WOODS: Synthetic challenge and real-world datasets. We use **train-domain validation** and **test-domain validation** for our synthetic challenge datasets driven by correlation shift. We use **train-domain validation** and **oracle train-domain validation** for our real-world datasets which are likely driven by other kinds of shifts.

### G.2   Model selection for subpopulation shifts

Model selection in subpopulation shift dataset is a much simpler endeavor because access to domains is not restricted. We define the two model selection strategies for our real-world datasets as follows.

**Average domain validation**   We split all domains into training, validation and testing splits. The training splits of that dataset is used to train the model. We choose the model that gets the best average validation performance on all domains. We report the worst testing split performance of the chosen model.

**Worst-domain validation**   We split all domains into training, validation and testing splits. The training splits of that dataset is used to train the model. We choose the model that gets the best worst domain validation performance. We report the worst testing split performance of the chosen model.

