# OpenReview forum: "WOODS: Benchmarks for Out-of-Distribution Generalization in Time Series"
_TMLR — Accepted by TMLR_

### Review · Reviewer_k3SH · 2023-05-04

**Summary Of Contributions:**

The paper addresses the underexplored area of Out-of-Distribution (OOD) generalization in time series tasks. The authors present WOODS, a benchmark consisting of 10 time series datasets, covering a wide range of data modalities, such as videos, brain recordings, and smart device sensory signals. They develop a systematic framework for evaluating time series datasets and algorithms and adapt existing OOD generalization algorithms for time series tasks. Extensive experiments show that there is a significant room for improvement in empirical risk minimization and OOD generalization algorithms on these datasets, highlighting the unique challenges posed by time series tasks.


**Audience:**

Yes

**Broader Impact Concerns:**

The paper does not have serious ethical concerns.

**Claims And Evidence:**

Yes

**Requested Changes:**

While the paper is already making valuable contributions to the field, exploring large-scale pretraining could potentially enhance its impact. In particular, it would be interesting to investigate the impact of large-scale pretraining on time series tasks to determine if it can help address OOD generalization challenges. Comparing pretrained and non-pretrained models would provide valuable insights. In general, the paper is already in a good shape.


**Strengths And Weaknesses:**

Strength:

1. The problem addressed by this work is interesting and important. In particular, the paper explores the underexplored area of OOD generalization in time series tasks, which is crucial for various applications such as computational medicine, natural sciences and finance.
2. The WOODS benchmark is solid and contain data from diverse data modalities and a wide range of critical problems, offering researchers a versatile tool for evaluating algorithms.
3. The authors examine several important OOD baselines on the time series data, providing valuable insights and highlighting challenges in time series tasks.


Weaknesses:
1. The paper does not explore the potential benefits of large-scale pretraining for time series tasks, leaving it an open question for future research. For instance, in WILD 2.0, they include the results with foundation models.
2. It is interesting to see some baselines such as IRM perform worse than ERM even in synthetic and toy setting, as it is specifically designed to address the spurious correlation. Is there any plausible explanation to this?

Overall, the paper does not have significant weaknesses and the contribution is solid.

---

> ### Author Response · Authors · 2023-06-07
> **Authors’ response to Reviewer k3SH**
>
>
> Thank you for taking the time and effort to review our manuscript and for sharing your perspective on the importance of time series OOD generalization. We appreciate your valuable feedback regarding large-scale pre-training and the need for clarification of our results. In the following paragraphs, we will address your concerns point-by-point. Our ultimate goal is to enhance the quality of our work, and we are more than willing to engage in continued discussion if any outstanding issues remain or arise.
>
> ### On large-scale pre-training for OOD generalization in time series
>
> We agree with the reviewer that large-scale pre-training holds promise for improving OOD generalization in time series. We base this belief on the reported zero-shot and few-shot performance of large open self-supervised vision and language models trained on web-scale data, as well as other controlled studies such as [1,3]. However, our understanding of the time series model pre-training landscape [2] has led us to the conclusion that the field is not yet mature enough to yield significant performance gains out of the box. This is primarily due to the absence of large-scale time series datasets in the current state of the field.
>
> That being said, transfer learning in time series has shown improvement when appropriate upstream pre-training datasets are used for downstream tasks [2]. However, the performance gains are highly dependent on the similarity between the upstream and downstream datasets. Given our focus on studying this behavior across a large set of diverse datasets with complex distribution shifts, finding combinations of beneficial upstream and downstream datasets would require a comprehensive and structured analysis to eliminate confounders and draw meaningful conclusions. We consider this topic to be outside the scope of the current work and leave it for future research (Note: the concept of large-scale pre-training for addressing distribution shifts was mentioned as an interesting research direction in the original Wilds paper [4] and further expanded upon in Wilds 2.0 [3].)
>
> ### ERM vs baselines results explanation
>
> While algorithms like IRM and VREx were specifically designed to address the failure of deep learning models due to spurious correlations, studies have shown [5] that they provide little to no generalization improvement over ERM on toy spurious correlation problems (e.g., ColoredMNIST) when model selection is not allowed using queries from the test domain.
>
> The explanation lies in the definition of spurious correlation and the design of toy datasets, such as ColoredMNIST, Spurious-Fourier. These datasets intentionally make attributes with spurious correlations (e.g., color of digits or spurious frequency) better predictors in the training domains. In our WOODS toy datasets, we set 80% and 90% correlation for spurious attributes and labels, while invariant attributes have a 75% correlation. Consequently, when comparing a perfectly invariant model (75% accuracy across all domains) and a spurious model (85% accuracy on training domains, 10% accuracy on the test domain) with train-domain validation, the spuriously correlated model obtains better train-domain performance. Selecting the invariant model only occurs if none of the models in the set leverages spurious correlation, which is rare in practice because of primacy bias of “easy” features. A more formal discussion on this topic can be found in Appendix G, which provides further justification for our model selection algorithms.
>
> ********************References********************
>
> [1] Kim, D., et al. (2022). A broad study of pre-training for domain generalization and adaptation. In *Computer Vision–ECCV 2022: 17th European Conference, Tel Aviv, Israel, October 23–27, 2022, Proceedings, Part XXXIII* (pp. 621–638).
>
> [2] Ma, Q. et al. (2023). A Survey on Time-Series Pre-Trained Models*. arXiv preprint arXiv:2305.10716*.
>
> [3] Shiori Sagawa, et al. (2022). Extending the WILDS Benchmark for Unsupervised Adaptation. In *International Conference on Learning Representations (ICLR)*.
>
> [4] Koh, Pang Wei, et al. (2021). WILDS: A Benchmark of in-the-Wild Distribution Shifts. In *International Conference on Machine Learning (ICML)*.
>
> [5] Ishaan Gulrajani, & David Lopez-Paz (2021). In Search of Lost Domain Generalization. In *International Conference on Learning Representations (ICLR)*.

---

### Review · Reviewer_7KNQ · 2023-05-08

**Summary Of Contributions:**

This paper proposed WOODS, a benchmark suite for out-of-distribution generalization research on time series datasets. WOODS consists of ten synthetic and real-world datasets in both classification and forecasting tasks. The authors first gave a formal definition of OOD generalization and then explained why OOD in time series is different and harder than traditional static setting. Next, they described the construction of each dataset. After that, authors used DomainBed suite to perform experiments on these datasets. The main conclusion is that OOD in time series remains a challenging problem and several OOD approaches do not significantly outperform ERM.

**Audience:**

Yes

**Claims And Evidence:**

Yes

**Requested Changes:**

1. Add definition to subpopulation shift.
2. Add more forecasting datasets.
3. Add more time series OOD algorithms.
4. Add more discussions as said in weakness section.
5. Add more backbones.

**Strengths And Weaknesses:**

### Strength

1. The topic is very necessary and interesting: OOD in time series remains an important and unsolved problem.
2. There are a lot of efforts in building such a complex and various benchmark, with benchmarking results and discussion.

### Weakness

1. The definition of subpopulation shift is missing in Section 2, but the results are just shown in experimental sections. You should add the definition in problem formulation.
2. The diversity of the datasets is not enough: there is only one forecasting dataset and the remaining nine datasets are all for classification. While time series classification is common and popular, the forecasting task is more unique in this domain. Therefore, I would encourage authors to add more forecasting datasets such as weather forecasting.
3. Authors only benchmarked existing general OOD algorithms while ignoring the advance of algorithms that are designed specifically for time series in OOD settings. I can provide several references (see below) and authors are encouraged to add more in their revision.
4. There should be more analysis towards the OOD generalization problem in time series rather than simply benchmarking algorithms. For instance, what makes it harder for time series in OOD setting? What can future research focus on?
5. Finally, the experiment section needs more details. For instance, what is the backbone of the experiments? As can be seen, there lacks a unique backbone in time series, not like ViT/ResNet in CV and BERT in NLP. So, the backbone matters in the results. Authors should clearly state the backbones in the paper and analyze the performance of different backbones. For instance, Transformers are known to work well in time series, so how about its performance in OOD settings?

Reference on time series OOD algorithms:

[1] Ragab, Mohamed, et al. "Conditional Contrastive Domain Generalization for Fault Diagnosis." IEEE Transactions on Instrumentation and Measurement 71 (2022): 1-12.

[2] Lu et al. "Out-of-distribution Representation Learning for Time Series Classification". ICLR 2023.

---

> ### Author Response · Authors · 2023-06-07
> **Authors’ response to Reviewer 7KNQ (Part 1/2)**
>
>
> Thank you for dedicating your time and effort to reviewing our manuscript. We also appreciate your recognition of the importance of our work and the extensive effort invested in this manuscript. Your valuable feedback on various aspects of our work is highly appreciated. In the following paragraphs, we will address your concerns point-by-point. Our ultimate goal is to enhance the quality of our work, and we are more than willing to engage in continued discussion if any outstanding issues remain or arise.
>
> ### Adding distribution shift definitions
>
> We have provided proper definitions for "domain generalization" and "subpopulation shift" distribution shifts in Section 2.1. Thank you for pointing out this omission.
>
> ### Adding forecasting datasets to WOODS
>
> We have included a new forecasting dataset with source domains, called **PedCount**, which has been adapted from [1]. Detailed information about this dataset can be found in Section 4.6 and Appendix C.11. The performance of forecasting baselines on PedCount is presented below and can also be found in Table 6 of the revised manuscript.
>
> ****Generalization gap (RMSE)****
>
> | Algorithm | ID | OOD | Gap |
> | --- | --- | --- | --- |
> | ERM | 99.1 (2.7) | 204.9 (11.4) | 105.8 |
>
> **Baseline performance (RMSE)**
>
> | Algorithm | Train-domain validation | Oracle train-domain validation |
> | --- | --- | --- |
> | ERM | 204.1 (11.4) | 223.2 (7.1) |
> | VREx | 201.6 (6.0) | 213.1 (3.1) |
> | GroupDRO | 243.2 (13.0) | 242.1 (9.9) |
> | IB-ERM | 213.1 (10.9) | 205.7 (11.3) |
>
> ### Adding time series specific OOD generalization baselines in WOODS
>
> We have added both proposed baselines, **Conditional Contrastive Domain Generalization (CCDG)** [2] and **Diversify** [3], to our set of baselines. The results are summarized below and can also be found in Table 3 and 4 of the revised manuscript. Thank you for bringing those works to our attention, we believe they complement well the existing set of baselines in WOODS.
>
> **************Train-domain validation**************
>
> | Algorithm\Dataset | Spurious-Fourier | TCMNIST-Source | TCMNIST-Time | CAP | SEDFx | PCL | LSA64 | HHAR |
> | --- | --- | --- | --- | --- | --- | --- | --- | --- |
> | CCDG [2] | 11.2 (0.7) | 10.1 (0.0) | 9.9 (0.1) | 61.7 (1.0) | 68.2 (0.6) | 64.3 (0.2) | 53.0 (1.2) | 84.7 (0.7) |
> | Diversify [3] | 12.2 (0.9) | 9.9 (0.1) | 10.2 (0.0) | 57.4 (1.9) | 76.9 (0.1) | 64.4 (0.4) | 48.3 (1.9) | 85.2 (0.7) |
>
> **************Oracle Train-domain validation**************
>
> | Algorithm\Dataset | Spurious-Fourier | TCMNIST-Source | TCMNIST-Time | CAP | SEDFx | PCL | LSA64 | HHAR |
> | --- | --- | --- | --- | --- | --- | --- | --- | --- |
> | CCDG [2] | 50.6 (0.2) | 49.8 (0.3) | 49.2 (0.3) | 63.1 (0.6) | 69.2 (0.4) | 64.4 (0.5) | 56.0 (1.6) | 85.8 (0.3) |
> | Diversify [3] | 67.0 (3.2) | 29.1 (1.4) | 29.8 (2.1) | 62.3 (1.1) | 77.2 (0.1) | 64.2 (0.4) | 50.1 (1.1) | 86.7 (0.6) |
>
> CCDG and Diversify demonstrates average performance on the domain generalization WOODS datasets. However, Diversify stands out with impressive results specifically on the SEDFx datasets. As concurrent research to this work, Diversify serves as further evidence that the field of OOD generalization in time series is actively evolving and progressing.
>
> ### Adding discussions and future research directions
>
> We have added Section 6.3 to discuss the challenges ahead in time series OOD generalization tasks and highlight potential avenues for future research directions.

---

> > ### Author Response · Authors · 2023-06-07
> > **Authors’ response to Reviewer 7KNQ (Part 2/2)**
> >
> >
> > ### On the backbone-oriented study of OOD generalization in time series
> >
> > We acknowledge that the backbone information should be included in the main body of the manuscript. To address this, we have aggregated the backbone descriptions from the Appendix into Table 2, which we have added to the main body.
> >
> > Our main focus in this work is to investigate the performance of OOD algorithms. For this, we require a fair way to compare methodologies, which we achieve by using the same backbone across baselines for a given dataset. While studying which backbone performs best for OOD generalization in time series could yield interesting results, we leave this as a topic for future work. We have made our codebase fully open-source and documented to facilitate further analysis in these areas. Our intention is to minimize barriers to entry into this field and encourage the exploration of interesting directions.
> >
> > ### Minor rectification
> >
> > > […] authors used DomainBed suite to perform experiments on these datasets.
> > >
> >
> > We would like to clarify that while our benchmarking suite is inspired by DomainBed in its evaluation protocol, it has been developed from scratch to be highly flexible and tailored to the unique challenges of time series datasets. This flexibility allows us to address various scenarios such as single/multiple predictions, source/time domains, and classification/forecasting tasks effectively.
> >
> > ********************Reference********************
> >
> > [1] Godahewa, R., et al. (2021). Monash Time Series Forecasting Archive. In *Proceedings of the Neural Information Processing Systems Track on Datasets and Benchmarks*.
> >
> > [2] Ragab, M., et al. (2022). Conditional Contrastive Domain Generalization for Fault Diagnosis*. IEEE Transactions on Instrumentation and Measurement, 71, 1-12.*
> >
> > [3] Wang Lu, et al. (2023). Out-of-distribution Representation Learning for Time Series Classification. In *International Conference on Learning Representations (ICLR)*.

---

> > > ### Comment · Reviewer_7KNQ · 2023-06-08
> > > **Thank you for your hard work**
> > >
> > > I would like to thank the authors for their hard work in adding experiments, algorithms, backbones, and discussions in this new version. This version looks good and I'm leaning towards acceptance. This paper is solid.
> > >
> > > Some minor concerns:
> > > - Figure 1 is not updated: since you added another forecasting dataset, figure 1 should also be updated to include it.
> > > - In page 3, when introducing the definitions of DG and subpopulation shift, you'd better include some references.
> > > - In table 3, why do most of the existing DG algorithms fail to outperform the simple baseline ERM? Any thoughts on that? Could it be due to imperfect hyperparameter tuning or something else?
> > >
> > > An even more minor comment:
> > > - The citation style should be updated: you should use "abc et al. (2023)" (\citet) and "(abc et al. 2023)" (\citep) better. Now you just mixed them.

---

> > > > ### Author Response · Authors · 2023-06-13
> > > > **Follow-up to Reviewer 7KNQ**
> > > >
> > > > We sincerely appreciate your recognition of the efforts we have put into improving our work, as well as your continued work on providing valuable feedback. In the following paragraphs, we address your remaining concerns and explain the revisions made to the manuscript.
> > > >
> > > > ### Updating Figure 1
> > > >
> > > > Thank you for bringing to our attention the need to update Figure 1. We have now included the PedCount dataset in Figure 1 of the revised manuscript.
> > > >
> > > > ### Problem formulation references to prior work
> > > >
> > > > We recognize the importance of providing references when introducing the definitions of DG and subpopulation shift. This has been addressed. Thank you for pointing this out.
> > > >
> > > > ### On the performance of DG algorithms
> > > >
> > > > Our evaluation reveals that, under realistic performance evaluations, most OOD generalization algorithms do not outperform ERM. While this observation may appear surprising, similar conclusions have been drawn in the context of non-time series domains [1,2].
> > > >
> > > > In our study, we adhere to the gold standard of evaluating OOD generalization algorithms through hyperparameter optimization. It is worth noting that our hyperparameter search algorithm is specifically created to minimize any potential bias introduced by human intervention which can inadvertently incorporate knowledge of the test domain into the training process and lead to artificially inflated performance.
> > > >
> > > > Although most existing OOD generalization algorithms perform worse than ERM, we demonstrate significant improvements achievable through our proposed generalization gaps, as illustrated in Table 1. This is encouraging for future research directions in the domain of OOD generalization for time series.
> > > >
> > > > ### Citation style
> > > >
> > > > We have made our citation style consistent throughout the manuscript. The updated citation style improves the overall presentation quality of the manuscript, thank you for highlighting this issue.
> > > >
> > > > ******************References******************
> > > >
> > > > [1] Ishaan Gulrajani, & David Lopez-Paz (2021). In Search of Lost Domain Generalization. In *International Conference on Learning Representations (ICLR)*.
> > > >
> > > > [2] Ye, N., et al. (2022). Ood-bench: Quantifying and understanding two dimensions of out-of-distribution generalization. In *Proceedings of the IEEE/CVF Conference on Computer Vision and Pattern Recognition* (pp. 7947–7958).

---

> > > > > ### Comment · Reviewer_7KNQ · 2023-06-14
> > > > > **Thanks for your updating**
> > > > >
> > > > > I would like to thank the authors for their continuous update of this work to make it stronger. I have no more questions. To my view, this paper can be accepted.

---

### Review · Reviewer_QoQM · 2023-05-24

**Summary Of Contributions:**

This paper presents a set of 10 datasets on time series. It includes 7 real-world datasets and 3 synthetically generated datasets. The authors also report on the existing OOD generalization methods to demonstrate the effect of distribution shifts.

**Audience:**

Yes

**Claims And Evidence:**

Yes

**Requested Changes:**

Please refer to the above comments.

**Strengths And Weaknesses:**

Pros:
I agree with the authors that there has been limited exposure for generalization under distribution shift for time-series-based tasks. Introducing there datasets are useful.


Cons:
1. My main concern is regarding the quality of the distribution shifts, especially on real-world datasets.
In Table 1, we observe that the effect of distribution shift (i.e., the gap between in-domain and OOD performance) is not significant (except for the LSA64 dataset), even for ERM. Further, the performance improves as we apply more sophisticated techniques (Table 2: Train-domain validation).
Finally, as we apply early stopping based on the validation set, the performance significantly improves, particularly for synthetic datasets.

2. Please provide a more detailed summary table for the descriptions of datasets such as the number of instances, shape of inputs etc.

Also, I believe that such work of introducing new datasets, without technical novelty, should be submitted to a more dataset-focused track.

---

> ### Author Response · Authors · 2023-06-07
> **Authors’ response to Reviewer QoQM (Part 1/2)**
>
> Thank you for dedicating your time and effort to reviewing our manuscript. We greatly appreciate your recognition of the value of our work, which aims to shed light on out-of-distribution (OOD) generalization in time series. We also appreciate your valuable feedback regarding the quality of the datasets and the presentation of our work. In the following paragraphs, we will address your concerns point-by-point. Our ultimate goal is to enhance the quality of our work, and we are more than willing to engage in further discussion if any outstanding issues remain or arise.
>
> ### On the quality of the presented distribution shifts
>
> First, regarding the absolute values of the generalization gap, we would like to emphasize that the WOODS datasets we have presented fall well within the range of what is considered a failure due to distribution shift in the field. Notable works, such as Wilds [1] (refer to Table 1) and Wild-Time [2] (refer to Table 1), provide concrete examples of standard generalization gap values.
>
> Second, the performance of the baseline algorithms we have presented should not contribute to our assessment of whether problematic distribution shifts exist within the datasets. OOD generalization algorithms serve as a solution to this problem. Moreover, as shown by our experimentation the WOODS datasets are not solved by current baseline algorithms. To illustrate this, consider real-world domain generalization classification datasets where the best baseline improves the performance of ERM by only ~1.1% on average. This leaves a significant generalization gap that needs to be addressed.
>
> Third, oracle model selection results, such as Oracle Train-Domain Validation and Test-Domain Validation, should not be employed to evaluate the generalization gap. These results cheat on the test distribution and should not be misconstrued as an indication of the current capabilities of OOD generalization methods. To draw an analogy with classical machine learning, using oracle model selection would be akin to performing hyperparameter selection and early stopping on the test dataset, i.e., the dataset used to report performance. Oracle model selection is solely used as a proxy to evaluate the absolute performance of methodologies, recognizing that it is impossible in real-world scenarios.
>
> Lastly, we have invested considerable thought into how we measure the generalization gap of our datasets. The methodology we have chosen aims to provide the closest possible measure that captures the impact of distribution shift on the performance of our models, while minimizing confounding factors. Appendix E provides detailed insights into our motivation and methodology.
>
> ### Adding detailed summary of technical dataset details
>
> As per your suggestion, we have consolidated the technical descriptions from the Appendix into Table 2, which we have now included in the main body of the manuscript. We appreciate your idea for this improvement.

---

> > ### Author Response · Authors · 2023-06-07
> > **Authors’ response to Reviewer QoQM (Part 2/2)**
> >
> > ### On the relevance to the track
> >
> > According to the **[acceptance criteria](https://jmlr.org/tmlr/acceptance-criteria.html)** of this TMLR track, we think this manuscript deserves its place. We explain how we fit the criteria below.
> >
> > > Are the claims made in the submission supported by accurate, convincing, and clear evidence?
> > >
> >
> > We have curated a collection of 10 (now 11) time series datasets with significant distribution shifts, encompassing diverse data modalities such as videos, brain recordings, and smart device sensory signals. We claim to have identified and demonstrated, through extensive experimentation, that the WOODS datasets exhibit problematic distribution shifts that remain an open problem to be solved, as none of the investigated baselines have adequately addressed the generalization issue. Our experimental approach follows the gold standard within the field of OOD generalization [4] for evaluating methodologies.
> >
> > > Would some individuals in TMLR's audience be interested in the findings of this paper?
> > >
> >
> > As you and other reviewers have acknowledged in your comments, the field has provided limited exposure to generalization under distribution shift for time-series-based tasks. Introducing the WOODS datasets is highly valuable to the field. We believe that a decent proportion of the TMLR audience would find the findings of this paper compelling. Our work facilitates easier entry into the study of OOD generalization in time series, as it eliminates the need to search for datasets to evaluate proposed methodologies, while providing an open source benchmarking suite that allows for standardized results.
> >
> > As a final note, it is worth mentioning that curated benchmarks of datasets studying OOD generalization have traditionally been part of the main track at major conferences [1,3,4]. We consider our work to fall under the same category of contributions.
> >
> > **References:**
> >
> > [1] Koh, Pang Wei, et al. (2021). WILDS: A Benchmark of in-the-Wild Distribution Shifts. In *International Conference on Machine Learning (ICML)*.
> >
> > [2] Yao, H., et al. (2022). Wild-time: A benchmark of in-the-wild distribution shift over time*. Advances in Neural Information Processing Systems, 35, 10309–10324.*
> >
> > [3] Shiori Sagawa, et al. (2022). Extending the WILDS Benchmark for Unsupervised Adaptation. In *International Conference on Learning Representations (ICLR)*.
> >
> > [4] Ishaan Gulrajani, & David Lopez-Paz (2021). In Search of Lost Domain Generalization. In *International Conference on Learning Representations (ICLR)*.

---

### Author Response · Authors · 2023-06-07
**General response**

We would like to express our gratitude to all the reviewers for their diligent efforts in reviewing our manuscript. We sincerely appreciate the valuable feedback provided in the initial reviews, and we have worked hard to address all the concerns raised during this review period. In this response, we will provide an overview of the updates made to the manuscript, and we will address each individual comment in direct replies to the reviews.

## Updates to the manuscript

### Increasing the representation of forecasting datasets in WOODS

We have added a forecasting dataset called **PedCount** to WOODS. The dataset is adapted from [1]. Detailed information about this dataset can be found in Section 4.6 and Appendix C.11 of the revised manuscript. We present the performance of forecasting baselines on the PedCount dataset in Table 6 of the revision.

### Increasing the coverage of WOODS’ baselines

We have added two additional baselines, **Conditional Contrastive Domain Generalization (CCDG)** [2] and **Diversify** [3], to our set of baselines. These algorithms complement the existing baselines in WOODS by covering a broader horizon of works in time series OOD generalization. The results are summarized in Table 3 and Table 4 of the revised manuscript.

### Expanded discussion on current state and future directions to OOD generalization in time series

We have added Section 6.3 discussing the challenges involved in time series OOD generalization tasks and provided potential avenues for future research directions.

### Writing and presentation improvements

- We have included proper definitions for "domain generalization" and "subpopulation shift" in Section 2.1.
- We have consolidated the technical descriptions and the backbone used for each dataset from the Appendix into the new Table 2, which we have added to the main body of the manuscript.
- We made minor formatting changes to stay within 12 pages (Figures 3 to 12 and Tables 3,4,5).

******References******

[1] Godahewa, R., et al. (2021). Monash Time Series Forecasting Archive. In *Proceedings of the Neural Information Processing Systems Track on Datasets and Benchmarks*.

[2] Ragab, M., et al. (2022). Conditional Contrastive Domain Generalization for Fault Diagnosis*. IEEE Transactions on Instrumentation and Measurement, 71, 1-12.*

[3] Wang Lu, et al. (2023). Out-of-distribution Representation Learning for Time Series Classification. In *International Conference on Learning Representations (ICLR)*.

---

### Decision · Action_Editors · 2023-07-08

**Recommendation:** Accept as is

**Comment:**

The reviewers identified a number of issues, which authors have addressed in the revision, including: adding new forecasting dataset, adding 2 new baseline methods, discussions of future work, clearly stated definitions of some terms, and other presentation issues.

One reviewer still had a concern "in Table 3 (Oracle train-domain validation), baseline models often significantly closed the performance gap between ID and OOD settings. However, I agree that one single model did not always achieve the best performance. Also, except LSA64 dataset, the performance gap between ID and OOD remains quite low." Responding, authors noted that: 1) the absolute values of the generalization gap are within the range considered as failures in the field; 2) the oracle performance listed Table 3 "cheats" by using the test data to select the best model, and thus should not be considered as the current capabilities, but rather an upper-bound.

The AE agrees with the authors, and also notes that benchmarks are necessary to ensure measurable progress in the field. As performance on a benchmark becomes saturated newer or larger benchmarks will be introduced.

The benchmark provides a common platform for evaluating OOD methods on a variety of time-series data, making an important contribution to field to ensure measurable progress for future research. Thus, the AE also recommends a "featured certification".

**Audience:**

The dataset will be interesting to researchers in time-series modeling and OOD. It will be a highly valuable resource as a benchmark for comparison of OOD methods to further research in this area.


**Claims And Evidence:**

The paper curates 11 time-series datasets with distribution shift as a benchmark for evaluation methods for OOD generalization in time-series. Extensive experiments are conducted, showing none of the considered baselines adequately addresses the generalization issue.